



# Cloud Regimes Over the Amazon Basin: Perspectives From the GoAmazon2014/5 Campaign

Scott E. Giangrande[1], Dié Wang[1], and David B. Mechem[2]

[1]Environmental and Climate Sciences Department, Brookhaven National Laboratory, Upton, NY, USA
[2]Department of Geography and Atmospheric Science, University of Kansas, Lawrence, Kansas, USA

*Correspondence to*: Scott E. Giangrande (sgrande@bnl.gov)

**Abstract.** Radiosonde launches collected during the GoAmazon2014/5 campaign are analyzed to identify the primary thermodynamic regimes accompanying different modes of convection over the Amazon. This analysis identifies five thermodynamic regimes that are consistent with traditional Amazon calendar definitions for seasonal shifts, which include a wet, transitional, and three dry-season clusters. A multisensor ground-based approach is used to project associated bulk cloud and precipitation properties onto these regimes to assess the propensity for each regime for characteristic cloud frequency, cloud types, and precipitation properties. Additional emphasis is given to those regimes that promote deep convective precipitation and organized convective systems. Overall, we find reduced cloud cover and precipitation rates to be associated with the driest regimes and those with the highest convective inhibition CIN. While approximately 15% of the dataset is designated as organized convection, these events are predominantly contained within transitional regime days.

## 1 Introduction

A primary source of uncertainty in global climate or earth system model (GCM, ESM) predictions of possible climate change is the representation of cloud processes and associated cloud feedbacks that regulate Earth's energy and water cycles (e.g., Klein and Del Genio, 2006; Del Genio, 2012). One explanation for continuing deficiencies in cloud–climate model process representations points to uncertainties in how deep convection is parameterized. Unfortunately, the assumptions underpinning the parameterizations are often poorly constrained by observations. Formulating well-behaved convective parameterizations necessitates routine cloud observations, married to their associated meso- and synoptic-scale controls, and collected over the variety of global convective regimes. Untangling these cloud–climate controls in ways suitable to ongoing model development demands long-term, multi-scale, multi-sensor observations that often require challenging instrument deployments to capture cloud and precipitation properties in remote and under-sampled global regimes (e.g., Louf et al., 2019).

As home to the largest tropical rainforest on the planet, the Amazon basin experiences prolific and diverse cloud conditions that vary according to pronounced changes in seasonal regimes. However, these clouds, regimes and their intensity are interconnected, with cloud properties (coverage, depth, precipitation) strongly influenced by (and influencing, via feedbacks) seasonal shifts in the thermodynamic forcing, as well as larger-scale atmospheric Hadley and Walker circulation variability



(e.g., Fu et al., 1999; Machado et al., 2004; Misra, 2008). Recently, the ongoing inability of GCM and weather prediction

models to represent aerosols, clouds and their interactions over this expansive tropical area motivated the 2-year US Department of Energy (DOE) Atmospheric Radiation Measurement (ARM) Observations and Modeling of the Green Ocean Amazon (GoAmazon2014/5) campaign (e.g., Martin et al., 2016; 2017). As part of this effort, ARM deployed its Mobile Facility (AMF; e.g., Miller et al., 2014) downstream of Manaus, Brazil in the central Amazon. The facility enabled capture of the thermodynamic state, aerosol, cloud and precipitation properties in this location, through the deployment of multiple

surface state and atmospheric profiling facilities (e.g., Mather and Voyles, 2013).

To isolate the potential controls of large-scale conditions on the clouds experienced over this region, we perform a cluster analysis on the routine radiosonde launches collected during the GoAmazon2014/5 campaign. A k-means clustering technique is employed to classify the primary thermodynamic regimes that are associated with the cloud observations over Manaus.

Conceptually, this technique follows previous tropical clustering efforts such as Pope et al. (2009a, b) to examine the variability found in the North Australia monsoonal seasons. Their motivations were to promote objective methods to identify key monsoonal changes, and establish periods favoring distinct cloud conditions to target global model evaluation and process characteristics (e.g., May and Ballinger, 2007). A similar opportunity is expected for Amazon studies, as several recent efforts (Marengo et al., 2017; Wright et al., 2017; Sena et al., 2018) illustrate the complex processes and possibly changing nature of

yearly transitions from dry and rainy seasons in the Amazon and its associated changes to cloud properties. The clustering approach may also yield an improved understanding of the relationship between the intraseasonal variability and the different Amazon convective regimes (Betts et al., 2002; Ghate and Kollias 2016). Moreover, there is continuing need to identify particular seasonal, environmental or aerosol controls on Amazon convection and its intensity (Greco et al., 1990; Williams et al., 2002; Alcântara et al., 2011; Fan et al., 2018; Wu and Lee, 2019; Rehbein et al., 2019).


For this study, the proposed regime segregations are projected onto the large-scale synoptic patterns, forcing datasets, and remote-sensing cloud/precipitation observations for the GoAmazon2014/5 campaign. These efforts are used to assess possible controls and convective-cloud predictors as related to i) the interpretation and consistency of these radiosonde clusters with previous wet/dry seasonal definitions for the Amazon, ii) bulk regime relationships to particular cloud presence/absence, the

iii) precipitation properties for these regimes to include diurnal cycles, and iv) the propensity for regimes to promote extremes in precipitation such as null-event days or mesoscale convective systems (MCSs, Houze, 2004). The GoAmazon2014/5 datasets are briefly described in section 2. The clustering algorithm, displays of the regimes according to thermodynamic variability, and additional methodology sensitivity testing are described in sections 2 and 3. Section 3 also explores the relationships between these regimes and overarching synoptic patterns, as well as area-averaged and observationally

constrained vertical profiles (e.g., horizontal moisture convergence) often used to force single-column models (SCMs). Summaries of cloud properties under these regimes are found in section 4. This includes discussion on the propensity for the





regimes to promote precipitation, and the likelihood of MCS events initiating nearby the campaign facilities. Finally, key findings for this study are summarized in section 5.


## 2 GoAmazon2014/5 Dataset and Processing Methods

Datasets for this study were collected by the U.S. DOE ARM facility during its "Observations and Modeling of the Green Ocean Amazon 2014–2015" campaign near Manaus, Brazil from January 2014 through December 2015 (herein,
GoAmazon2014/5 or MAO; Martin et al., 2016; 2017; Giangrande et al., 2017). The primary datasets were from the routine ARM radiosonde launches during the campaign at the "T3" main AMF field site downwind of the city of Manaus, Brazil and near Manacapuru, Brazil. These radiosondes provide the thermodynamic quantities of interest and act as basis for regime clustering methods (section 2.2).

## 2.1 ARM GoAmazon2014/5 Products and Datasets

Details on ARM radiosondes, their preprocessing and convective parameter estimates, follow previous ARM studies (e.g., Jensen et al., 2015). The quantities of interest for this study include estimates for the convective available potential energy (CAPE), the convective inhibition (CIN), the Relative Humidity (RH) at low- (surface to 3 km), mid- (3 km to 6 km) and high-
levels (above 6 km) of the atmosphere, the 0–5-km wind shear, the Level of Free Convection (LFC), and the 0–3-km Environmental Lapse Rate (ELR). The originating parcels for CAPE/CIN estimates are defined by the level of the maximum virtual temperature in the lowest kilometer. Thus, the standard calculations for CAPE and CIN represent the most buoyant parcel in the boundary layer (below 700 hPa) such that the reported values are comparable to 'most unstable CAPE/CIN' (herein, MUCAPE/MUCIN). Mixed-layer CAPE and CIN estimates (mean parcel properties over the lowest 500 m, which we
take to be representative of the mixed layer) were also computed for comparison.

Interpretations for the cloud properties associated with regime breakdowns (clusters) are supported by collocated instruments at the MAO site, as well as observationally constrained reanalysis datasets. For precipitation properties, surveillance S-band (3 GHz) radar observations were available to within 70 km of the MAO site as collected by the System for the Protection of
Amazonia (SIPAM) radar located near the city of Manaus (e.g., Ponta Pelada airport, Martin et al., 2016). These radar data were calibrated against satellite measurements, and subsequently gridded to a 2 km × 2 km horizontal grid at 2 km AGL (e.g., Schumacher and Funk, 2018).

Cluster routines incorporate only the morning (1200 UTC, 0800 local time) radiosondes that are launched in clear conditions.
Clear conditions are defined as having no rainfall at the MAO site according to rain gauge measurements to within an hour of





launch time. Confirmation of cloud-free conditions was also performed using SIPAM observations and manual checks for contaminated radiosondes. A motivation for using the morning radiosonde was to capture pre-convective cloud conditions prior to the daily transition from clear to shallow cumulus to deep convection, given the known diurnal precipitation cycle for Manaus that peaks near local noon (e.g., Giangrande et al., 2017). Additional concerns are that earlier (0600 UTC) or later

(1800 UTC) radiosonde launches are not representative of the pre-convective environment, and are more susceptible to existing clouds, overnight fog (e.g., Anber et al., 2015), or precipitation contamination. In total, 607 daily radiosondes from the campaign (out of 696, 12-UTC radiosondes in total) met these criteria, with 27 days removed due to missing radiosondes. Of the days flagged as contaminated or 'missing' at 1200 UTC, approximately 30-40 days were associated with radar-designated MCS passing over MAO (section 4).


Time-height (column) cloud properties are provided by a hybrid cloud radar / radar wind profiler (RWP) product developed during GoAmazon2014/5 (Giangrande et al., 2017; Feng and Giangrande, 2018). The product combines the ARM multi-sensor (e.g., cloud radar, lidar, ceilometer, radiometer) Active Remote Sensing of CLouds (ARSCL; Clothiaux et al., 2000) cloud boundary designations with collocated 1290 MHz ultra-high frequency (UHF) RWP measurements (e.g., Giangrande, 2018;

Wang et al., 2018), and gauge observations. The RWP bolsters the ARSCL cloud-boundary designation through deeper precipitating clouds that attenuate the cloud radar measurements of cloud echo top. A simple cloud type classification is performed following McFarlane et al. (2013) and Burleyson et al. (2015). Observed clouds are classified into seven categories according to the height of the cloud and cloud thickness (Supplemental Table S1). These seven cloud categories are 'shallow', 'congestus', 'deep convection', 'altocumulus', 'altostratus', 'cirrostratus/anvil', and 'cirrus'.


Large-scale synoptic perspectives on the regimes are obtained using reanalysis fields from ERA5 (Hersbach and Dee, 2016) and the ARM variational analysis product (ARM-VARANAL). The VARANAL is derived from ECMWF analysis fields and ARM observations during GoAmazon2014/15 using the constrained variational analysis method of Zhang and Lin (1997). The product is available at 3-hour intervals on a regular vertical grid of 25 hPa over a domain of ~110 km radius around the MAO

site (Xie et al., 2014; 2016). The product is also constrained by the domain-mean precipitation as observed by the SIPAM radar. Additional details on these products during GoAmazon2014/5 are found in Tang et al. (2016).

**2.2 *K*-means Clustering Methods**

Regime classification is accomplished using an open-source Scikit-learn's *k*-means algorithm applied to input radiosonde observations (toolkit from Pedregosa et al., 2011). The choice of *k*-means solutions over other configurations is done for simplicity and is consistent with previous radiosonde applications. While the sensitivity of proposed regime designations to different clustering approaches is not the subject of this study, applying alternate configurations did not alter relative breakdowns or composite interpretations.






A primary shortcoming when applying *k*-means clustering to these problems is that the number of clusters needs to be prescribed. One expectation for the Amazon is that three to four regimes account for the bulk seasonal thermodynamic variability: i) a 'wet' season regime typically defined as December through April, ii) a 'dry' season regime from June through September, and iii) one or two 'transitional' regimes associated with the months leading into the wet and dry regimes,

respectively. From sensitivity testing (see section 2.3), we establish the number of clusters at five (Figure 1; Herein, we use the terms 'cluster' and 'regime' interchangeably). Radiosonde temperature and wind information is input at 20 equally-spaced levels from 1000 hPa to 200 hPa, as similar to previous applications over North Australia (Pope et al., 2009a,b). This input resolution is coarser than the resolution of the ARM radiosondes (~2 hPa), and that of the 25-hPa VARANAL resolution. Additional tests (not shown) indicate that, for this particular case, the *k*-means solutions are insensitive to improvements in the

input radiosonde resolution, or input order. Although the authors prefer the solution that does not use normalized inputs (e.g., scaling to similar range, standard deviation), such inputs are common practice, and select consequences are discussed when these inputs result in divergent solutions.

Presenting cluster breakdowns according to calendar-based Amazon definitions for the wet, dry and transitional seasons

(Figure 1), the dry season months (Figure 1, bottom left panel) are predominantly associated with regimes 1-3. Traditional Amazon wet season months (Figure 1, top right panel) are associated with regimes 4 and 5, with negligible contributions from the remaining regimes. The ambiguous transitional season (calendar residual months) indicates contributions from all regimes, though skewed towards regimes 4 and 5.

In Figure 2, we plot the time-series of regime designations throughout the campaign (top panel), with the associated monthly breakdowns for the clusters (bottom panel). Qualitatively, the temporal coherency of the five-regime solution provides initial confidence in the appropriateness of these breakdowns. Instances of regimes 4 and 5 are aligned with classical transitional and wet season periods, respectively, with regime 4 periods adjacent to regime 5 and not sporadically distributed within other regimes. The remaining clusters are interwoven within Amazon dry season months. The observed cycling between dry season

clusters is of immediate interest, as this variability may be indicative of intraseasonal synoptic pattern phases in the dry season.

The specifics of the GoAmazon2014/5 campaign and its particular representativeness in the context of historical Amazon records should be considered when assessing cluster appropriateness. As summarized by Marengo et al. (2017), climatological wet season onset for Manaus based on rainfall records is typically mid-November (e.g., Liebmann and Marengo, 2001). Their

efforts indicate that rainfall trends and wet season onset measures such as outgoing longwave radiation indicators (e.g., Kousky, 1988) imply that the 2014-2015 wet season onset date occurred much later in the season (e.g., end of January, 2015). One explanation for the late onset, offered by Marengo et al. (2017), was that precipitation - the obvious indicator for wet-season onset - was heavily influenced by the strengthening of the Madden–Julian Oscillation (MJO; Madden and Julian, 1994) and





associated influences on Amazon rainfall. Based on cluster outcomes in Figure 2, we did not identify a prolonged cluster
arguably associated with a presumed 'wet season' condition (e.g., regime 5) until early December 2014. This coherent shift in
the frequency of radiosonde regime 5 designations coincides with an extended change-over in the upper-level winds, as also
shown in campaign thermodynamic summary plots (e.g., Fig. 2 from Giangrande et al., 2017). Nevertheless, we record multiple
instances of regime 5 as early as November 2014, coinciding with a pronounced dry-to-wet seasonal shift towards a deep-layer
profile moisture (RH, see also Fig. 2, Giangrande et al., 2017). As before, the motivation for such breakdowns is not to
'pinpoint' an exact rainy season onset date (e.g., first appearance of a given cluster), rather to identify atmospheric regimes
that may provide guidance towards subsets of attendant environmental conditions conducive to different bulk cloud properties.

## 2.3 Additional *k*-means Cluster Sensitivity Considerations

Establishing the number of clusters within *k*-means methods requires sensitivity testing. Too few clusters tends to
overgeneralize and produce overly large intra-cluster variability; too many clusters lead to difficulty in interpretation, because
there may be no physically meaningful distinction between clusters. Similar to justifications proposed by Pope et al. (2009a,
b), we are interested in regimes associated with significant radiosonde variability, and therein, potential relationships to cloud
variability. One criterion those authors recommended was that each cluster accounts for no less than 10% of the dataset. When
adopting this approach, Amazon breakdowns having more than five clusters generated additional clusters that accounted for
fewer than 10% of the days.

When considering a six-cluster solution (supplemental Figure S1), the solution further subdivided the three drier-season
regimes into four. However, the distinct separation between our wet (regime 5) and transitional (regime 4) clusters showed
little difference when the number of clusters was increased from five to six. To be discussed in section 3, the wet and
transitional regime separations predominantly differ from each other in their zonal/meridional wind structures. This does not
suggest that there are not specific differences depending on whether the transition is wet-to-dry and dry-to-wet, only that these
differences are not as pronounced as the drier intraseasonal shifts. In contrast, the four-cluster solution meets our basic criterion
for determining the number of clusters (Supplemental Figure S2). However, with only four clusters, the regime 4 and 5 clusters
are combined into a single, deep-moisture profile regime. Because of this, the authors settle on the five-cluster solution as it
maintains a separate transitional regime that the authors believe is consistent with the literature.

## 3 Thermodynamic and Large-Scale Interpretation of Amazon Regime Clusters

### 3.1 Composite Regime Thermodynamic Profiles and Parameter Displays





In Figure 3, we plot the composite radiosondes for all five regimes classified in the previous section. Shaded regions provide reference to composite radiosonde MUCAPE (red shading) and MUCIN (blue shading). Values reported on these images are

the median values of the MUCAPE/MUCIN calculated for each individual sounding. The probability density plots in Figure 4 report the median values, distribution, quartiles and $10_{th}/90_{th}$ percentile extremes for the convective parameters of interest estimated from the radiosondes. Differences in MUCAPE and MUCIN across the regimes are largely driven by differences in moisture rather than temperature, a result consistent with the understanding that horizontal temperature gradients over the tropics are small, and variability in tropical convection is associated with horizontal moisture gradients ("weak temperature

gradient approximation," Sobel et al., 2001). For all regimes, the standard deviations for MUCAPE and MUCIN parameters are similar (1100 J/kg and –15 J/kg, respectively). For other fields, the standard deviations vary with regime, with greater variability in the dry season than in the wet season. For example, standard deviation for wind shear is 4–6 m/s in the wet season versus 2–4 m/s in the dry season. For mixed-layer CIN values, median regime values shift to larger magnitudes (-33 J/kg for regime 5, to -85 J/kg for regime 1), however the relative distributions and regime rankings are similar. When considering

mixed-layer CAPE distributions, the values estimated for regime 1 (the highest MUCAPE regime) noticeably shift lower than the other regimes (median values dropping to 550 J/kg), with the remaining regimes having similar median mixed-layer CAPE values of approximately 1000 J/kg (similar relative rankings otherwise). This discrepancy in mixed-layer CAPE and more prohibitive mixed-layer CIN may explain the absence of deep convection under regime 1 conditions (section 4).

Temporal patterns for regime 5 align with calendar wet season definitions, while composite radiosonde and thermodynamic parameters point to deeper moisture conditions (Figure 3e, Figure 4). Overall, regime 5 is associated with reduced values for MUCAPE, but favorable MUCIN (i.e., less negative) to promote frequent convection (e.g., Giangrande et al., 2016). Regime 5 also records the lowest LFC and LCL heights, and reduced distribution variability therein. Where regime breakdowns differ from traditional Amazon ideas is with the frequency our methods define wet-to-dry season months such as April through June

as 'transitional' regime 4 (Figure 3b) periods. The most significant difference we observe between the regime 4 and 5 composites are associated with profile winds, which includes increased lower-level wind shear in regime 4 (Figure 4f). This particular separation for wet and transitional regimes as according to wind shifts is consistent with ideas of transpiration or shallow convection 'preconditioning' an eventual wet season onset (e.g., Wright et al., 2017), e.g., favorable moisture conditions precede deeper cloud formation prior to regional scale wind shifts lending to wet season onset. However, this

explanation would not apply for the reciprocal wet-to-dry transitional periods. Nevertheless, this dry-to-wet transition may bear some resemblance to the moistening and associated cumulus and congestus that occur as the MJO over the tropical western Pacific transitions from suppressed to active conditions (e.g., Johnson et al., 1999; Benedict and Randall, 2007; Mechem and Oberthaler, 2012, Zermeño–Díaz et al., 2015). Finally, while the differences in bulk wind shear are interesting between regimes 4 and 5, the magnitude of these differences are modest (to within 5 m/s). However, differences in mean shear may be indicative

of differences in updraft structure (upright vs. tilted), convective cold pool circulations, and overall organization (e.g., Rotunno et al., 1988; Parker and Johnson, 2000; Weisman and Rotunno 2004) during regime 4.





Previous Amazon studies suggest that the dry-to-wet season transitional periods (e.g., September through November) are more conducive to storm electrification than wet-to-dry transitional periods (e.g., Williams et al., 2002). This clustering solution
does not distinguish differences between these periods (here, 'dry season' as traditionally defined, from June through September). Although the separations for regimes 1 (extreme dry) and 5 (extreme wet) are robust to our input tests, when $k$-means methods use normalized inputs, this change realigns five-cluster solutions towards 'pre' and 'post' dry season states (Supplemental Figure S3). While the authors do not pursue such solutions, one suggestion is that increasing the relative weight of the wind field inputs may differentiate 'transitional' periods. In our supplemental images, we provide composite properties
for pre- (March through May) and post- (September through November) dry season regime 4 instances (supplemental Figure S4). Current regime 4 solutions exhibit enhanced MUCAPE for soundings collected during dry-to-wet periods that suggests those times as more conducive for vigorous updrafts (median MUCAPE values greater by ~700 J/kg).

The remaining clusters are associated with months traditionally classified as the Amazon dry season. Shifts between the drier
season clusters are attributed to radiosonde mid-to-upper level moisture, with only minor controls associated with shifts in winds. Regime 1 is the least-frequently observed for the Amazon campaign, but the most significant outlier in terms of thermodynamic parameters (e.g., Figure 4). Regime 1 is also associated with the driest overall profile conditions (at low- and mid-levels), the lowest mixed-layer CAPE, the highest LFC and most prohibitive MUCIN conditions. Regime 3 favors humid conditions at the low-to-mid levels when compared to regimes 1 and 2, and modest mid-to-upper level humidity. These
conditions may increase the frequency to initiate deep convection and/or sustain detrained ice particles to enable stratiform processes. As widespread stratiform precipitation and MCSs have been reported also within the dry season (e.g., Wang et al., 2018; 2019), section 4 explores which dry season regime or regimes may favor MCS.

### 3.2 Large-Scale Synoptic Conditions Projected into these Regimes


In Figure 5, we plot the composite large-scale synoptic patterns, means of the 1000-hPa geopotential height and wind field from the ERA5, projected into each regime. For the wet regime (regime 5), the composites show land-ocean contrasts, and composites carry strong impressions of the Chaco low over the continent (and/or Bolivian high at the upper levels). Signatures of the Bolivian high are also viewed through the deep layer of prevailing southerly winds over the MAO site that is exclusive
to regime 5 composites (Figure 3e). Unlike other composites, regime 5 also suggests 1000-hPa flows providing moisture convergence into the Amazon basin originating from the tropical belt (northern tropical Atlantic, e.g., Drumond et al., 2014), and composite westerly wind components over the MAO T3 site. While the 1200 UTC regime thermodynamic profiles did not indicate a pronounced difference between regimes 4 and 5 moisture characteristics, ERA5 composites suggest that regime 4 conditions are associated with different sources of moisture, with winds over the Amazon basin shifting towards drier easterly
zonal 1000-hPa flows. One interpretation for the regime 4/5 shift as from large-scale composites may be interconnected to the





South Atlantic Convergence Zone (SACZ) positioning/strength and its influences on the Amazon basin during the wet season (e.g., Carvalho et al., 2004). Drier season regimes have transitioned to southerly low-level flow suggestive of drier, colder air reaching the central Amazon. These patterns vary according to the positioning and strength of offshore features that, in turn, funnel increasingly drier, colder air from the southeast (e.g., tropical South Atlantic; Drumond et al., 2014).


GoAmazon2014/5 recorded one complete transition from the dry season to the wet season as viewable by the current designations. In Figure 6, we plot the composite 1000-hPa patterns associated with regime 5, with each panel corresponding to a different monthly composite between October and January. Noting that few radiosondes of regime 5 were recorded for October, composite ERA5 maps suggest large-scale trends and flow patterns were reminiscent of regime 4 (e.g., transitional)

composites (Figure 5d), and with weak indications for a continental surface low pressure or moisture inbound from southward latitudes. December composite patterns, in contrast, better reflect parent regime 5 composite behaviors (e.g., Figure 5e), that shift towards significant westerly composite low-level flow and low pressure / SACZ patterns by January. Westerly shifts in the central Amazon rainy seasons have been previously discussed as promoting a moist troposphere and frequent (albeit, not necessarily more intense) convection if compared to easterly flow regimes near the beginning of the rainy season (e.g., Betts

et al., 2002; Cifelli et al., 2002; Peterson et al., 2002).

To further explore attendant large-scale conditions and regime transitions, we plot composite daily projections of horizontal moisture advection and vertical velocity from the VARANAL product (Figure 7). Estimated horizontal advection of moisture (e.g., $-V*\nabla q$ ; V is horizontal wind vector, $q$ is water vapor mixing ratio; top row, green shading) is highest (positive) at the

lower levels for the regime 4 and 5 clusters, and maximized at the lowest levels below 700-hPa around the 1200 UTC radiosonde launch time (dashed line). In terms of large-scale vertical velocity $w$ (bottom row), note that $w$ fields are constrained by the domain-mean precipitation (assimilated SIPAM observations). Specifically, the strength of vertical motion and/or diabatic heating is proportional to the precipitation rates used in the analysis (e.g., Xie et al., 2014). Regimes with higher precipitation rates will indicate stronger ascending motion associated with greater diabatic heating during the afternoon

precipitation periods. Interestingly, the large-scale $w$ patterns during the morning hours are similar between regimes 2 through 5. Similarly, each regime indicates large-scale subsidence above 600-hPa that peaks around radiosonde launch time. However, regime 1 is an outlier and suggests substantial large-scale subsidence (above 600-hPa) and the weakest lower-level upwards motion around the morning radiosonde.

Finally, we isolate the variational analysis profiles corresponding to the pre-convective radiosonde launches by plotting median profiles and $10_{th}/90_{th}$ percentile values at 1200 UTC (Figure 8). Regimes 4 and 5 share similar characteristics and enhanced moisture advection (lower levels) and larger-scale $w$ in the mean and extremes ($90_{th}$ percentile). Regime 4 also displays stronger upward motions from near the surface to 650-hPa, and stronger extremes in $w$ from ~750-hPa upward. Since 1200 UTC is prior to significant domain-mean precipitation (section 4.2), these enhancements in regime 4 motions are not influenced by





precipitation constraints. Similarly, moist regimes lack the extreme negative (dry) moisture advection (10th percentile

properties) found in regimes 1–3.

**4 Regime Cloud and Precipitation Summaries, Likelihood for Precipitation Extremes**


**4.1 Cloud Frequency**

Cumulative cloud frequency and diurnal summaries that correspond to Figure 7 examples are plotted in Figure 9. The

characteristics are in-line with monthly breakdowns previously available for the GoAmazon2014/5 campaign as reported by

Collow et al. (2016). In Figure 10, we plot the frequency of specific cloud types for the periods following 1200 UTC

(radiosonde launch) to 0000 UTC, to include the relative frequency of null conditions over the site. For values reported in

Figure 10, multiple cloud layers can be identified in the same column; therefore, individual cloud types and null conditions do

not add up to 100%.

Cloud properties in Figures 9 and 10 indicate regime 1 is least favorable for cloud coverage (total, or daytime hours following

the radiosondes). This is consistent with the least-favorable 1200 UTC convective parameters, moisture advection and

subsidence as discussed by previous sections, as well as GoAmazon2014/5 dry-season studies on precipitation controls (e.g.,

Ghate and Kollias, 2016). During GoAmazon2014/5, regime 1 was the only regime where a majority of the daytime hours

over the site were not populated with clouds (e.g., Figure 10b). When clouds were present, the most frequent cloud type was

shallow cumulus ('shallow'). Upper-level cirrus clouds occupy a substantial fraction of the cloud observations under all

regimes, and are the second-most frequent clouds observed for regime 1 conditions. Presumably, the prevalence of cirrus in

regime 1 is attributable to cirrus generated remotely then being advected over the site. Interestingly, there is an absence of

cirrus and other cloud types in the periods around the 1200 UTC radiosonde launch (all regimes, Figure 9). This provides

confidence in our choice of 1200 UTC radiosonde for regime classifications that are not contaminated by clouds. All regimes

suggest large-scale subsidence at upper levels around 1200 UTC (e.g., Figure 7), which may explain the absence of cirrus.

The drier season cloud summaries in Figures 9 and 10 indicate increasing cloudiness from regimes 1 to 3, with cloud frequency

positively associated with reduced MUCIN (lower MUCAPE) and higher column RH. Dry-season cloud frequency (regimes

1-3), to include mid (congestus)-to-upper level (anvil, to include widespread/deep stratiform shields) cloud frequency, is

significantly lower than observed for regimes 4 and 5 (Figure 9g). Among drier season regimes, regime 3 conditions are most

conducive to clouds, although the relative cloud frequency breakdowns are similar-scaled to the cloud types in regime 2 (e.g.,

Figure 10). Moreover, diurnal cycles indicate that relative contributions from congestus are mostly absent from regime 1-3

mid-morning to afternoon periods (e.g., bimodal), and the increase in frequency between regimes 2 and 3 is attributed to

©️ Author(s) 2020. CC BY 4.0 License.





enhanced shallow (echo tops < 3 km) and deeper (isolated) convection (echo tops > 8 km). There is weak evidence of overnight
precipitating clouds during the dry season (e.g., Ghate and Kollias 2016), observed during the relatively moist regime 3.

MAO clouds are most frequently observed during the moist regimes (regimes 4 and 5), with increases in frequency attributed
to contributions from all cloud types. Regime 5 indicates the highest frequency for shallow to mid-level clouds (e.g., 'shallow',
'congestus', and 'alto'), and the highest frequency overall (e.g., Figure 9g). Diurnal plots suggest a gradual daytime shallow-
to-deep cloud transition for regimes 4 and 5, consistent with previous arguments for increased water vapor in the lower
troposphere as the primary factor responsible for triggering this transition (e.g., Ghate and Kollias 2016). Interestingly, the
bulk timing of this transition is potentially contingent on the regime, as this is apparently occurring later in the day according
to regime 5 composites. One explanation for the delayed timing is that this transition may be slowed by the reduced incident
solar radiation associated with more frequent shallow clouds under regime 5 conditions (Figure 9g). Variations in shallow-to-
deep timing are also consistent with differences in surface energy balance partitioning, which are a strong function of soil
moisture (e.g., Findell and Eltahir 2003a, b; Jones and Brunsell, 2009). Higher soil moisture values in the wet regime favor a
partitioning of the surface net radiation toward more latent than sensible heat flux (i.e., smaller Bowen ratio). This partitioning
leads to a moister boundary layer, but weaker generation of turbulent boundary-layer growth that should foster a slower
transition. Even in a tropical rainforest, the importance of moisture availability has been shown to have a large impact on
Bowen ratio (Gerken et al., 2018), suggesting this as a possible mechanism for modulating the onset of deep convection.

Regime 5 indicates a trimodal distribution of convective clouds, as observed in previous tropical studies (e.g., Johnson et al.,
1999). Over the tropical oceans, the congestus mode is associated with a mid-level stable layer near the melting (0°C) level
(e.g., Johnson et al., 1999; Jenson and Del Genio 2006). This is thought to arise from radiative interactions accompanying
intrusions of dry air from poleward latitudes (e.g., Mapes and Zuidema 1996; Redelsperger et al., 2002; Pakula and Stevens,
2009), or melting processes in organized stratiform precipitation (Mapes and Houze, 1995), though recent findings argue that
the melting mechanism is not essential to creating the stable layer (Nuijens and Emanuel 2018). How these two possible
mechanisms explain the presence of the congestus mode across the different Amazon regimes is not obvious. Regimes 1 and
2 are characterized by dry-air intrusions from poleward latitudes, yet exhibit the lowest frequency of congestus; this indicates
that other factors are strongly suppressing the vertical development of congestus and cumulonimbus. The higher frequency for
congestus during regimes 4 and 5 is accompanied by a greater incidence of organized convection (section 4.3); this suggests
the possibility of the stratiform-cooling mechanism. To complicate matters, only the composite soundings for regimes 2 and 5
(Figure 3) exhibit indications of a mid-level stable layer (~700–550 hPa).

Finally, bulk cloud characteristics are similar between regimes 4 and 5 during the morning to afternoon hours (Figure 10).
However, an important shift in cloud properties under regime 5 is observed during the pre-radiosonde (overnight) periods,
with regime 5 associated with more frequent congestus. From such depictions, it is unclear whether this shift in overnight





cloudiness in regime 5 is associated with more frequent or resilient congestus, or possible contributions from MCS. As discussed below, MCSs and/or radar-based indicators for widespread precipitation are more frequent for regime 4. This argues

that the increase is attributed to additional / resilient congestus, and this explanation is consistent with the modest upper (anvil) peak for regime 4 and prominent congestus peak observed for regime 5.

## 4.2 Differences in Precipitation Behavior Across Regimes

Model evaluation often benefits from precipitation constraints that include comparisons to the diurnal cycle and other precipitation properties. In Figure 11, we plot the diurnal cycle of precipitation from the domain-mean precipitation rate used to constrain the 3-hourly VARANAL products, contingent on the regime-events having measurable precipitation. For these breakdowns, precipitation rate (in mm/hr) is based on SIPAM estimates for the domain within the 110-km radius of MAO site. The dotted lines on Figure 11 correspond to the domain-mean values, and the shading indicates a 1-sigma standard deviation

for regime events. These standard deviations indicate the event-to-event variability; however, precipitation rates estimated by radar may carry at minimum 30% uncertainty (e.g., bias, or fractional root-mean-square error) owing to miscalibration or other factors (e.g., Xie et al., 2014; Giangrande et al., 2014).

For VARANAL-scale products, the MAO location favors a pronounced daytime diurnal cycle (Figure 11), with peak occurring

after local noon (e.g., 1800 UTC). The well-behaved diurnal cycle is consistent with climatologies over land from the Tropical Rainfall Measurement Mission (TRMM; Nesbitt and Zipser 2003; Yang and Smith 2006; Hirose et al., 2008), but this behavior may be fortuitous, since complex land surface cover, topography, or river / sea-breeze controls influence precipitation measurements in other parts of the Amazon basin (e.g., Burleyson et al., 2016; Machado et al., 2018). The cloudiest times over the MAO column do not perfectly align with domain-mean precipitation properties, but the most frequent clouds we observe

are typically near 1800 UTC (e.g., Figures 9, 11). Still, there are important shifts between various regimes. For example, regime 5 domain-mean precipitation skews higher than the other regimes from 2100 UTC into the overnight hours and associated with an increased MAO column cloudiness (e.g., Figure 9e). Overall, moist regimes favor more intense rainfall rates, with the highest rainfall rates observed in regime 4, followed by regime 5. Although fewer clouds, smaller total convective area, and lower-relative domain rainfall rates are observed during the drier season, the individual convective events

(updrafts, precipitation) can be quite strong (Giangrande et al., 2016; Machado et al., 2018). This is evident by the relatively high domain rainfall rates that are observed for regimes 2 and 3 for days when precipitation is recorded.

In Figure 12, we plot distributions for the maximum daily radar echo area after 1200 UTC (i.e., largest continuous area from any single radar scan) occupied by various thresholds for the reflectivity factor, as proxies for deep convective core area

coverage ($Z > 40$ dBZ) and widespread rainfall area coverage ($Z > 20$ dBZ). Thus, this measurement is a daily reference to the largest individual cell (any time), not a measurement for the total 'convective' area occupied by cells. Previous studies



including Giangrande et al. (2016) and Machado et al. (2018) have indicated that rainy seasons favor larger total convective area coverage. In terms of allowance for singular deeper convective cores (Figure 12a), it is not surprising that regime 4 (e.g., transitional) is associated with the largest convective cells, as based on higher expectations for MCS. In terms of convective

core properties associated with $Z > 40$ dBZ behaviors, multiple drier season distributions share comparable behaviors as to regime 5. This is consistent with suggestions that the dry season also promotes isolated, intense convection.

Nevertheless, regimes 4 and 5 favor a substantially wider distribution of widespread precipitation coverage (e.g., Figure 12b) as compared to the drier regimes. An increase in widespread precipitation coverage ($Z > 20$ dBZ) is consistent with the

arguments for more ubiquitous weak convection and/or MCS having trailing stratiform anvils (e.g., Romatschke and Houze, 2010). Interestingly, this may be interpreted as weaker cells/precipitation winning out over less frequent, but stronger cells. This is suggested as responsible for the reduced domain-mean precipitation rates compared to regime 2 (Figure 11 reflects only contributions from precipitation events). This is consistent with regime 3 as associated with additional congestus and/or periphery stratiform precipitation, enabled through reduced MUCIN and greater humidity above 600-hPa.


### 4.3 Radar-based Null Event or MCS Event Frequency

In addition to compositing clouds by regime, we explore a simple Bayesian approach to query the likelihood a particular regime promotes different precipitation modes, information that is highly useful for convective parameterization and predictive efforts.

If convection initiates for a given regime, what is the likelihood that the convection is nonprecipitating (e.g., defined by a minimal area of $Z > 20$ dBZ of $< 200$ km$^2$), isolated, or develops to a widespread precipitation event? In Figure 13, we break down the likelihood that precipitation events observed during GoAmazon2014/5 fall under nonprecipitating (NULL), isolated precipitating convection (ISO), and wide deeper convective (WDC) events. Among those WDC events, we identify those events having mature-stage MCS characteristics. For these mature MCS definitions, we follow the guidelines established in

Houze et al. (2015) and Feng et al. (2018), where MCS are defined as having continuous 40 dBZ radar echo area exceeding 1000 km$^2$, with a continuous shield of 20 dBZ radar echo areas exceeding 10000 km$^2$. WDC events are defined as the precipitation events having a continuous, widespread shield of 20 dBZ echo exceeding 10000 km$^2$. For simplicity, ISO events are defined as the events that did not fall within NULL or WDC categories (i.e., NULL + ISO + WDC = total events). For the analysis in Figure 13, 595 of the 607 rain-free radiosondes days were also well-observed by the SIPAM.


Overall, NULL precipitation days are rare, accounting for less than 4% of our two-year record (Figure 13, Table S2). NULL events were predominantly designated during the driest regimes, with regimes 1 and 2 accounting for 20 of the 23 (87%) instances. WDC events account for approximately 21% of the dataset, and commonly observed for regimes 4 and 5 (approximately 81%). Subsampling those WDC events, radar-based MCS events are relatively uncommon, accounting for

approximately 8% of the dataset. The majority of these MCS events were observed during the moist regimes (regimes 4 and 5





accounting for > 70% of the events), with approximately half of the MCS observed during regime 4 (Figure 13). For completeness, the number of MCSs during GoAmazon2014/5 was approximately double, as we ignore radar-based MCS that produced rainfall over the MAO site at the time of radiosonde launch. Additional manual inspection of the WDC events also reveals that one-third of WDC events shared MCS-like characteristics that fell short of study thresholds. Thus, potentially 20%

of the campaign period was associated with MCS, although only half are considered for our analysis. Similarly, MCS designations are subjective, and we anticipate inconsistencies between this accounting and satellite tracking (e.g., Rehbein et al., 2019). One final consideration is that MCSs do not need to initiate locally (e.g., within the SIPAM radar domain ~ 500 km) to meet our radar-based definitions. We have inspected radar and satellite observations for 44/47 MCS events to manually identify MCS from our criteria that initiated to distances >500 km upstream, then propagated over the site. Supplemental Table

S2 identifies two MCS categories, 'propagating', and 'local', as reminiscent of previous Amazon studies (e.g., Greco et al., 1990). By our breakdowns, MCS during the drier season are predominantly 'propagating' events, while moist regimes include contributions from both MCS categories.

As the regime most associated with mature MCS events, in Figures S5 and S6 we plot composite radiosonde and parameter

distributions (MUCAPE, MUCIN) for regime 4 'nonMCS', 'local' (13 events) and 'propagating' events (7 events). In Figure 14, we plot a similar MCS breakdown for 1200 UTC horizontal moisture advection and $w$ from VARANAL. Overall, we do not observe a significant difference between the composite properties among MCS and nonMCS events within regime 4. Similarities between MCS and nonMCS events are also reflected in the 1200 UTC variational forcing composites (Figure 14), with local MCS and nonMCS events reflecting comparable mean conditions. 'Propagating' MCS events are less representative

of composite behaviors and suggest weaker thermodynamic conditions with the most favorable large-scale controls. However, these large-scale moisture/velocity enhancements are modest (e.g., vertical velocity increase of 2.5-to-5 hPa/h).

**5 Summary**


To inform on the potential controls for clouds experienced over the Amazon basin, a cluster analysis was performed on routine radiosondes launched during GoAmazon2014/5. We identified five primary thermodynamic regimes and explored these states in the context of traditional Amazon definitions, composite large-scale synoptic patterns, and model forcing datasets. Column and scanning radar observations were projected into these states, highlighting the propensities for each state to promote

different cloud types, frequencies, and changes to precipitation. Emphasis was given to intra-regime conditions associated with organized convection in the transitional regime (regime 4) most favorable to MCS. A summary of the findings is as follows:

- $k$-means clustering of the 1200 UTC radiosonde datasets yields five primary clusters that correspond with Amazon wet, transitional and dry season cloud regimes. The three drier season regimes relate different states of mid-to-upper





level moisture associated with the strength of similar large-scale features that advect colder/drier air into the Amazon basin. The wet to transitional seasons exhibit similar deep moisture thermodynamic profiles, with regime 5 associated with evidence of moisture advection into the Amazon basin from the tropical belt.

- GoAmazon2014/5 cloud frequencies, cloud types and precipitation properties for the five regimes correspond well to bulk changes in the large-scale vertical air motion, moisture advection, local radiosonde thermodynamic composite

profile and convective parameter shifts. Most regimes favor frequent clouds and intense precipitation during the early afternoon hours (after 1600 UTC), with precipitation following a single-peak diurnal cycle.

- The moist regimes were associated with modest MUCAPE, reduced MUCIN and higher humidity at all levels. The latter two controls are those suggested as most favorable in the Amazon for more frequent clouds, deeper convection, and widespread stratiform precipitation. Regimes 4 and 5 also suggest prominent shallow-to-deep cloud transitioning

(with trimodal cloud profile behaviors observed in regime 5), with the timing of these transitions potentially contingent on the regime (e.g., later in the day under regime 5).

- The drier regimes reflect reduced column cloud frequency, bimodal instead of trimodal distributions in vertical profiles of cloud frequency, an absence of mid-level cloud contributions and shallow-to-deep transition signatures, and rainfall properties attributed to weak or isolated (infrequent) deep convection. Although convection is frequently

observed during all regimes, dry-season regimes are those attributed with less frequent clouds and rare Amazon NULL precipitation events.

- When precipitation is observed, SIPAM radar designations indicate most convection is isolated deeper convective cells. Approximately 10-20% of the convection observed over MAO was associated with MCS during this deployment. These MCSs were most frequently observed over MAO under moist profile conditions (regimes 4 and

5), with approximately half of the daytime (1200 UTC to 0000 UTC) and well-defined MCSs observed during GoAmazon2014/5 within regime 4 periods. Approximately half of the well-defined MCSs that passed over the site fell outside of the typical diurnal cycle and/or were not associated with regime classifications.

- When considering regime 4 favorability for deep convective events, it is suggested that intra-regime (pre- and post-dry season months) variability may account for shifts in favorability for enhanced storm updrafts and/or

electrification. However, this study did not identify shifts in composite thermodynamic profiles or convective parameter distributions between MCS and nonMCS conditions. Additional checks of the large-scale synoptic patterns and forcing datasets under MCS and nonMCS conditions indicate that 'propagating' MCSs may favor an enhancement in the large-scale vertical air velocity (2.5-5 -hPa/h) and moisture tendencies during pre-convective windows that offsets weaker local thermodynamic environments. However, these factors were arguably less important

when compared to overall regime 4 proclivity for MCS.



**Data Availability**

All ARM datastream to include VARANAL, ARSCL, SONDE and other PI datasets used in this study can be downloaded at http://www.arm.gov and are associated with several "value added product" VAP streams and GoAmazon2014/5 PI datasets. Python machine learning codes were provided by Scikit-learn, as from Pedregosa et al., (2011). ERA5 reanalysis products (production) are available at: https://www.ecmwf.int/en/newsletter/147/news/era5-reanalysis-production, as from Hersbach and Dee, (2016).


**Author contributions**. SEG, DW and DM designed the research; SEG and DW performed research; SEG and DM wrote the paper.


**Acknowledgements**

This study was supported by the U.S. Department of Energy (DOE) Atmospheric System Research (ASR) Program and the Climate Model Development and Validation (CMDV) program. This paper has been authored by employees of Brookhaven Science Associates, LLC, under contract DE-SC0012704 with the U.S. DOE. The publisher by accepting the paper for

publication acknowledges that the United States Government retains a nonexclusive, paid-up, irrevocable, worldwide license to publish or reproduce the published form of this paper, or allow others to do so, for United States Government purposes. Co-author Mechem was funded by U.S. Department of Energy Atmospheric Systems Research Grant DE-SC0016522. The authors would also like to thank Luiz Machado (INPE), Ernani de Lima Nascimento (UFSM), Jiwen Fan (PNNL) and Andreas Prein (NCAR) for helpful comments and discussion.

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






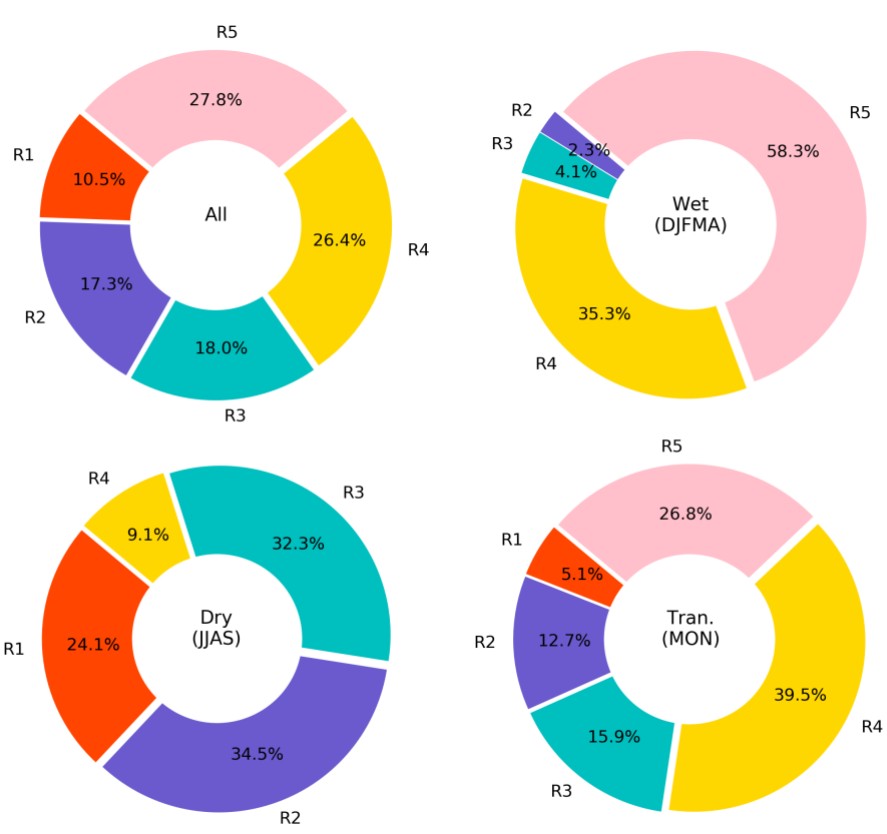

**Figure 1. Breakdowns for the frequency to observe regime clusters (regimes 1 through 5 marked as R1 through R5) for the GoAmazon2014/5 radiosonde dataset (1200 UTC), as well as breakdowns for wet season (Dec., Jan., Feb., Mar., Apr.), dry season (Jun., Jul., Aug., Sep.), and transitional season (May, Oct., Nov.) radiosondes.**









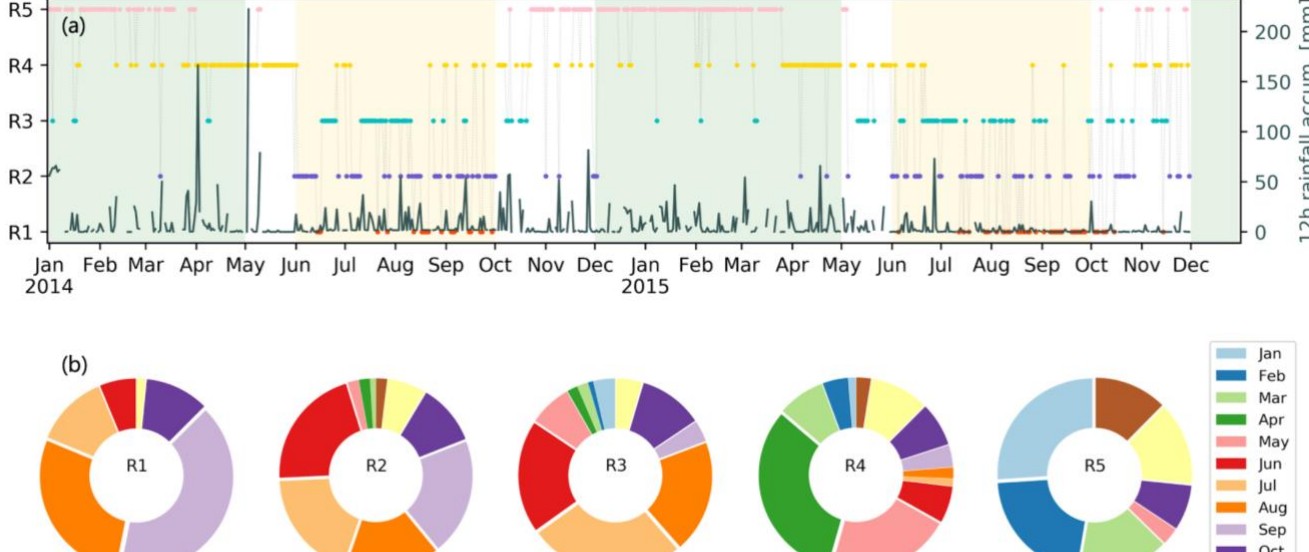

**Figure 2. (a) Time series for Amazon regime cluster results with corresponding 12h (1200 UTC - 00 UTC) rainfall accumulation (from the MAO rain gauge). The green shading indicates the wet seasons and the yellow shading indicates the dry seasons according to calendar definition; (b) Relative breakdown for the frequency of each regime according to month.**









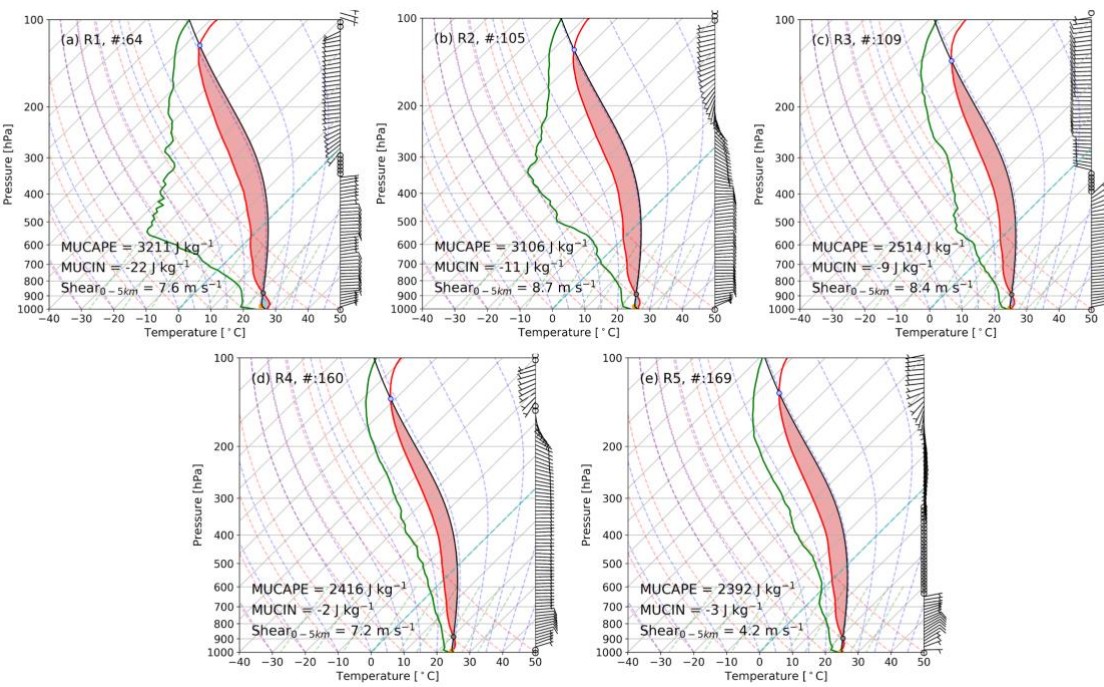

**Figure 3. Composite 1200 UTC radiosondes for each regime. MUCAPE, MUCIN, and wind shear (surface to 5 km) parameters report regime-median values.**







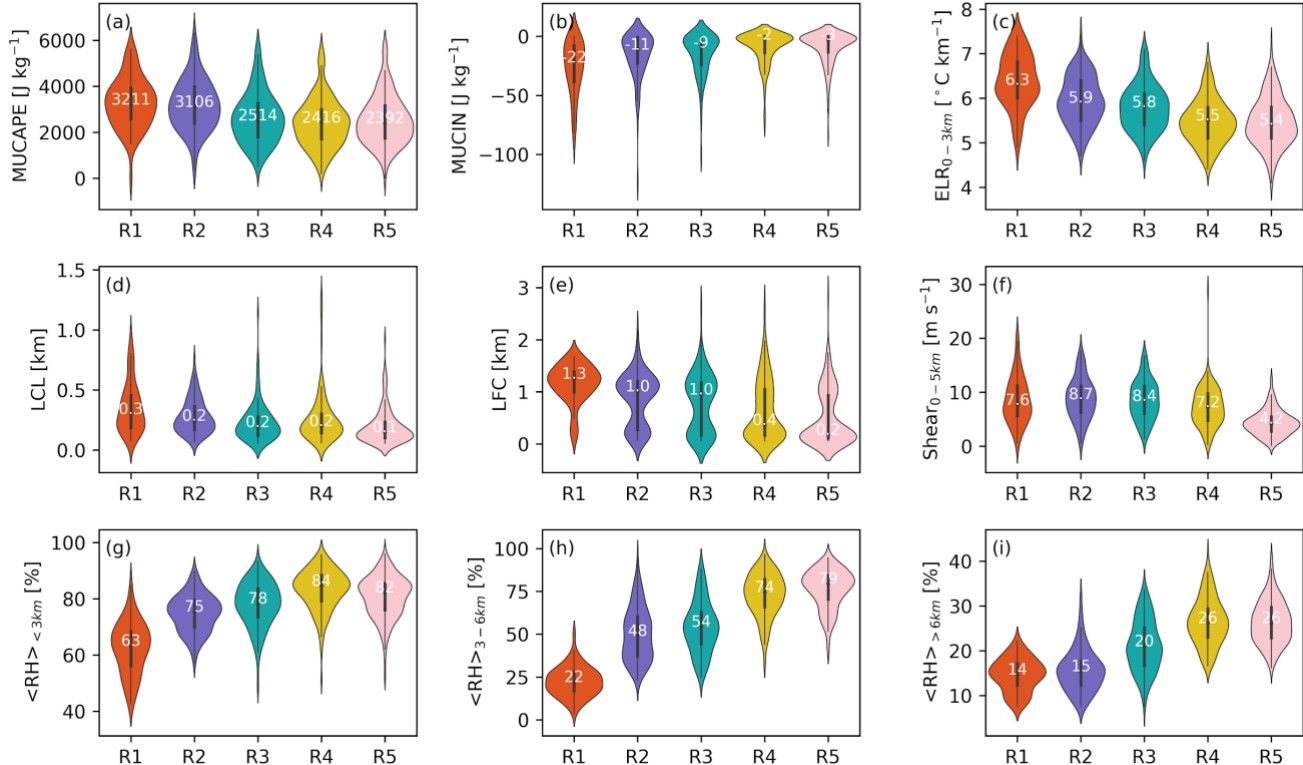

**Figure 4.** Shaded probability density plots for select thermodynamic quantities of interest estimated from the 1200 UTC radiosonde in each Amazon regime. The median values for each regime distribution are reported on each violin (white text).


12000





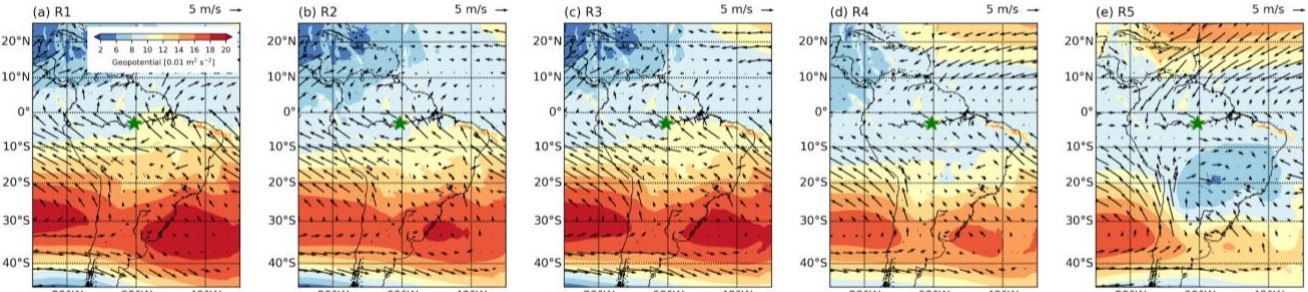

**Figure 5. Composite large-scale synoptic patterns (geopotential heights in color [0.01 m2 s-2] and horizontal winds) projected into each regime, as from ERA5 for the 1000-hPa level. The green star indicates the ARM MAO T3 site.**






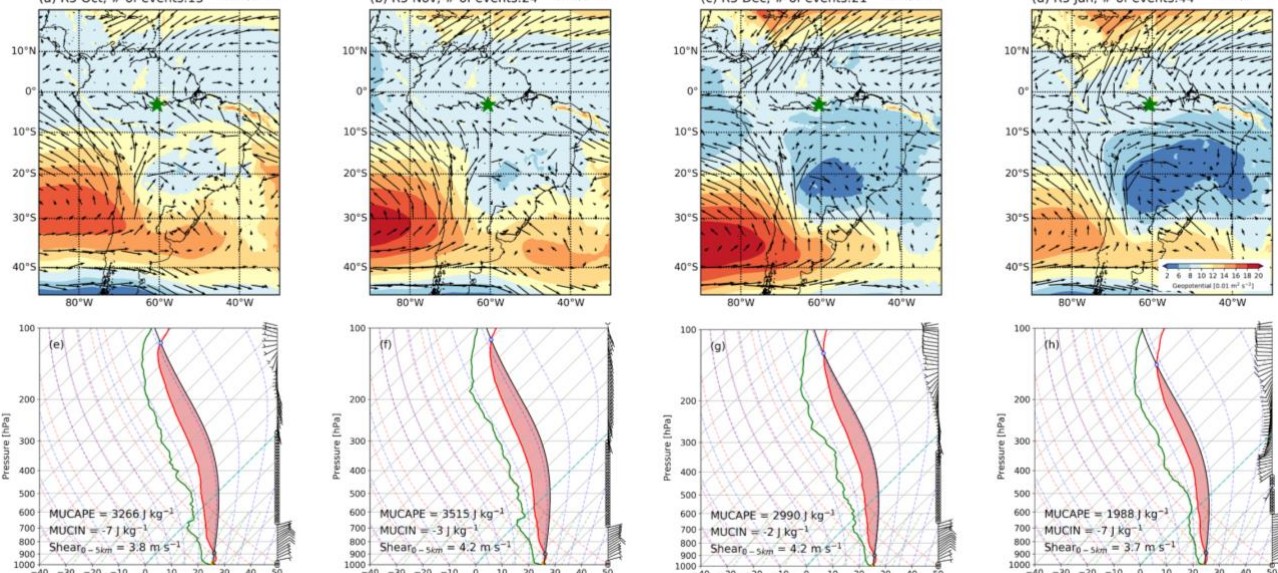

**Figure 6. Composite monthly large-scale synoptic patterns at 1000 hPa (following Figure 5) and radiosondes, associated with regime 5. Plots correspond left-to-right to (a) October, (b) November, (c) December, and (d) January.**







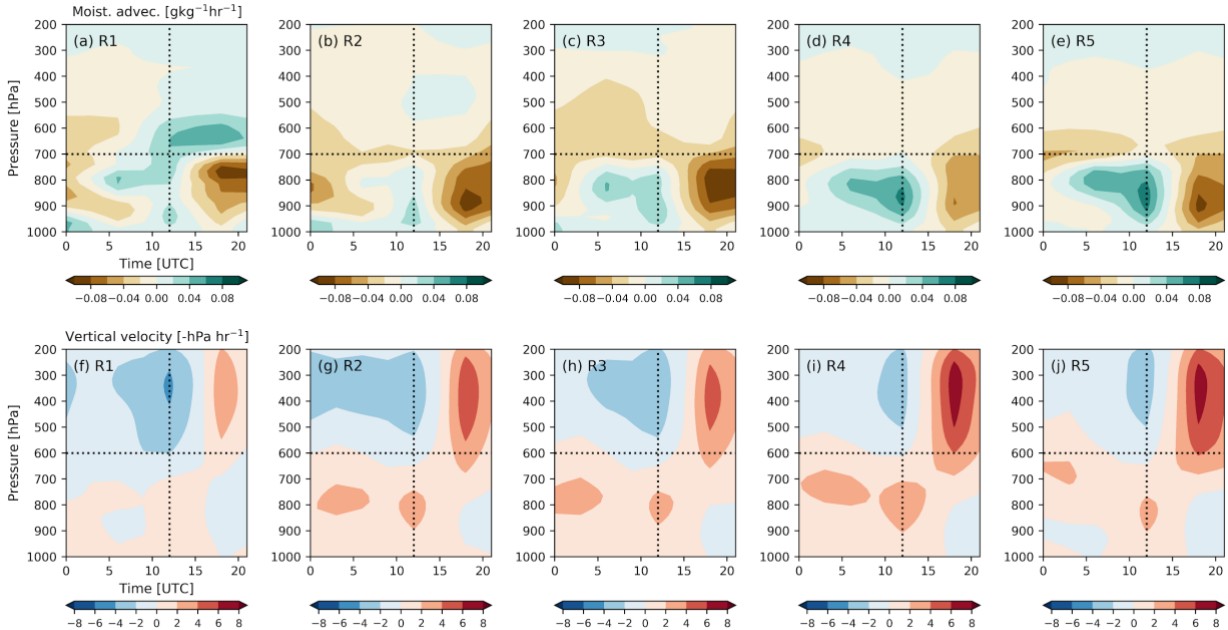

**Figure 7. Composite diurnal (UTC) large-scale SCM variational forcing dataset (VARANAL) fields for (a-e) regime breakdowns of the horizontal moisture advection (green = positive moisture advection), and (f-j) large-scale background vertical velocity (red = upward vertical motion). 1200 UTC columns and 600-hPa/700-hPa levels are highlighted as dotted lines.**











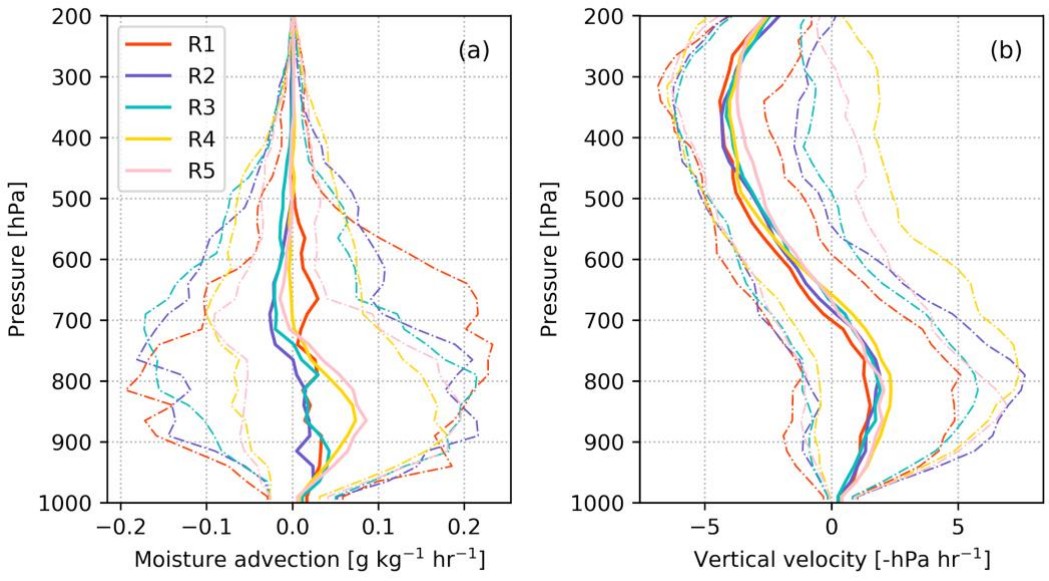

**Figure 8. Median profiles (thick solid lines) of (a) horizontal moisture advection and (b) large-scale background vertical velocity (positive value = upward motion) for each regime at 1200 UTC. The 10th and 90th percentile ranges for the variational analysis fields are represented by the dashed lines.**






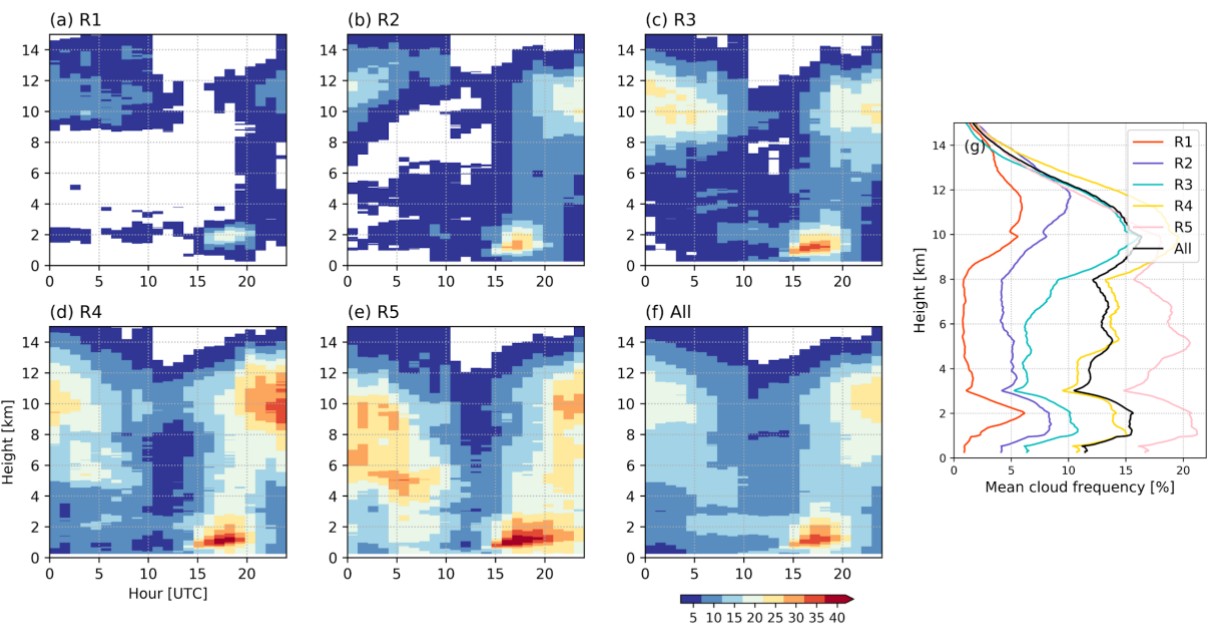

**Figure 9.** The diurnal cycle of hour-mean cloud frequency (when cloud coverage > 2%) as a function of height for each regime (a-f),
as according to a multi-instrument cloud profiling retrieval. The mean 1h cloud frequency profiles are shown in (g).











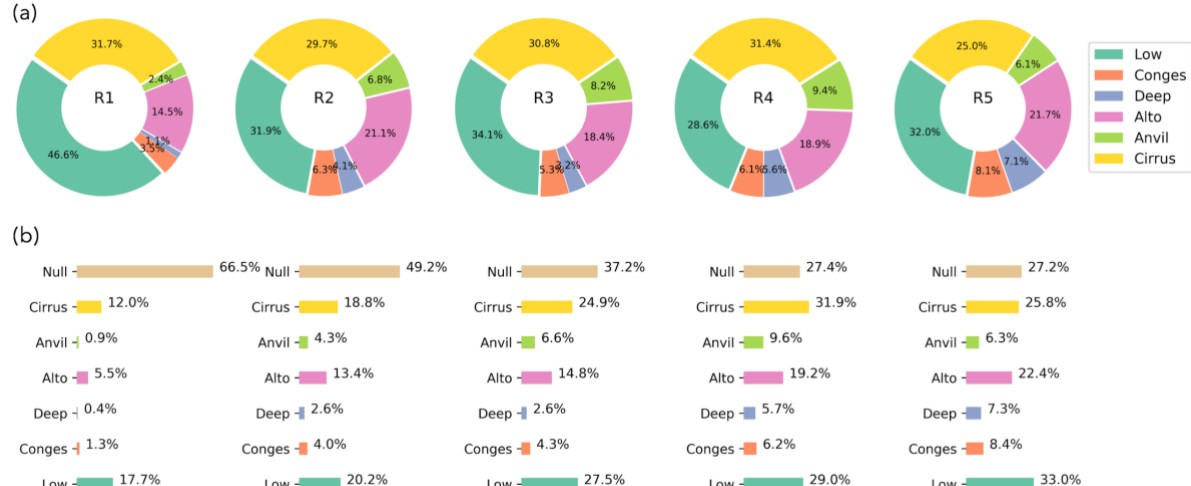

**Figure 10. (a) Relative frequency of occurrence for specific cloud types in the column above the ARM MAO T3 site for regime periods between 1200 UTC and 0000 UTC, and (b) percentages when compared to cloud-free conditions.**





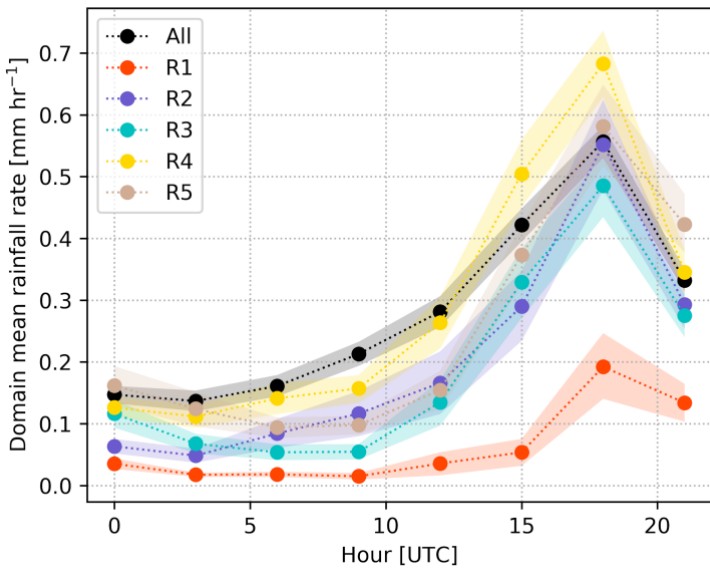

**Figure 11. Domain-mean precipitation rate (for events with measurable precipitation) from the SIPAM radar to within a 110 km**
**radius of the MAO site. The dotted lines report the dataset mean values, and the shading is 1-sigma standard deviation.**






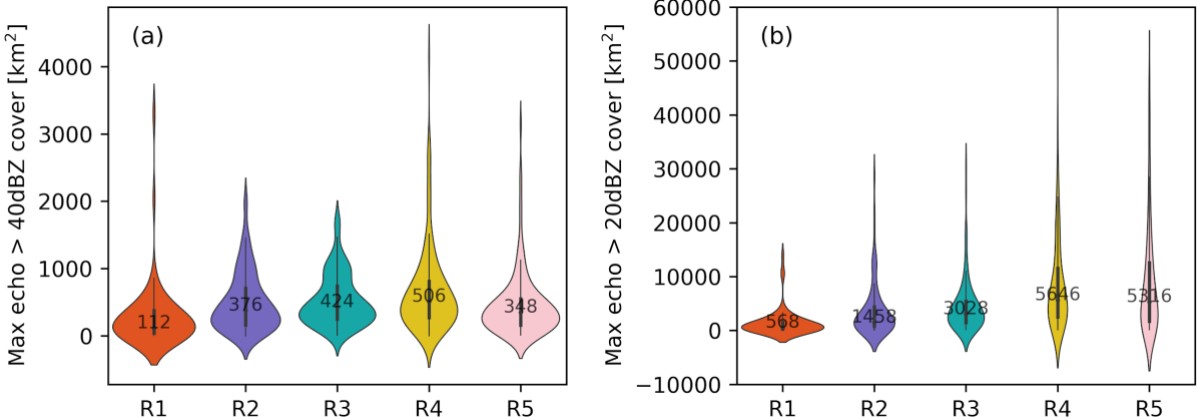

**Figure 12. As in Figure 4, the maximum contiguous 2 km CAPPI radar echo coverage [in km2] for any radar scan within a regime day that is occupied by radar echoes exceeding an intensity (a) Z > 40 dBZ, or (b) Z > 20 dBZ, for hours between 1200 UTC and 0000 UTC that day.**





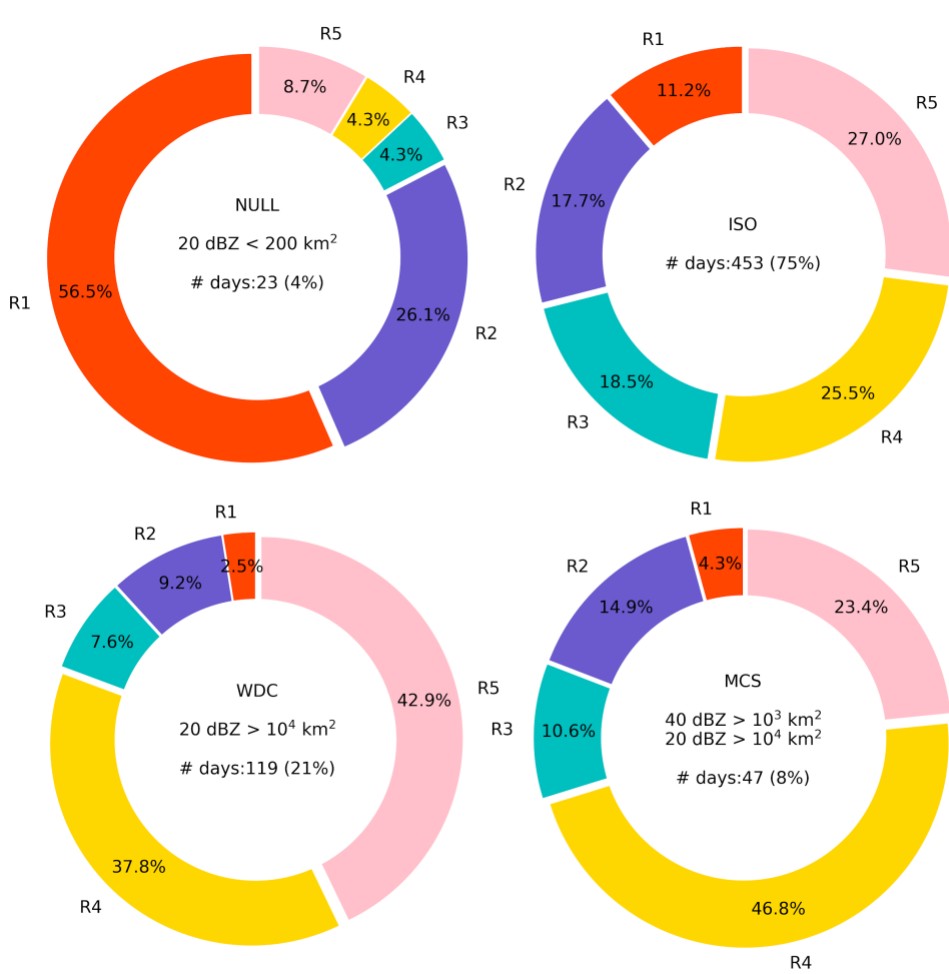

**Figure 13.** As in previous frequency plots, but for the percentage of (top left) NULL, (top right) ISOlated, (bottom left) wide deep convection (WDC) and (bottom right) MCS days associated with each regime cluster.





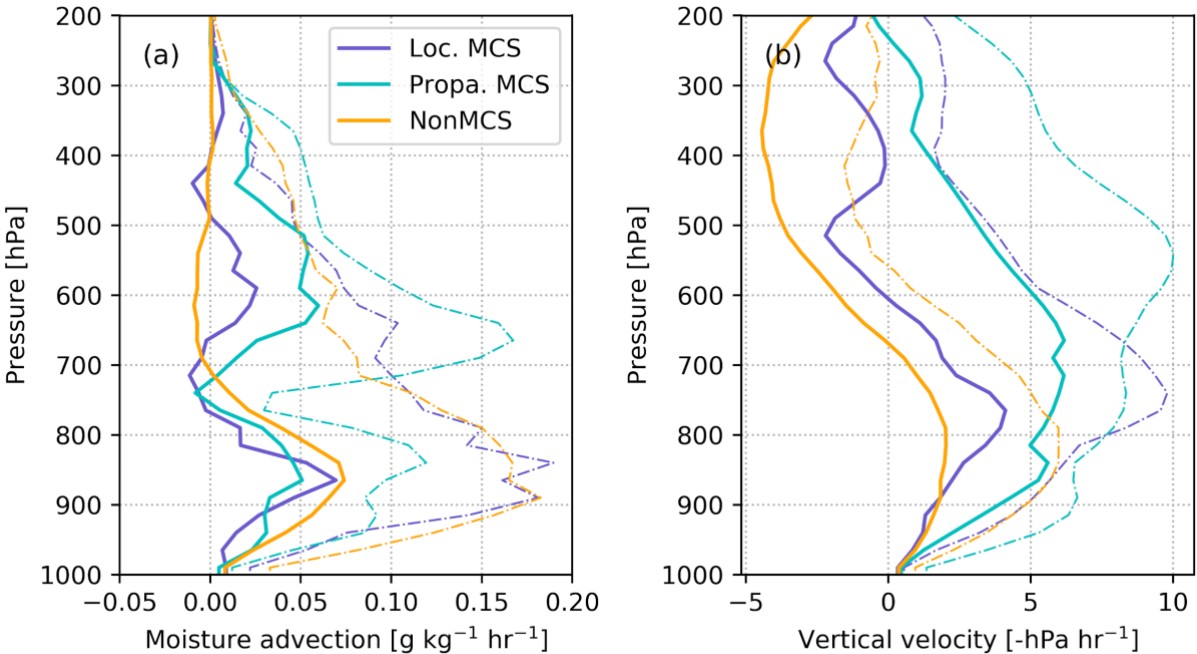

**Figure 14. Variational forcing profiles at 1200 UTC for nonMCS, local MCS, and propagating MCS cases with rain rate less than 1.5 mm/hr. Profiles correspond to the regime 4 conditions. Solid lines are median profile values and dashed lines are the 95th percentile values.**