# Peer review of "Cloud Regimes Over the Amazon Basin: Perspectives From the GoAmazon2014/5 Campaign"

_Atmospheric Chemistry and Physics, 2020_

## Referee Comment (RC1) · David Adams (Referee) · 9 Mar 2020

Review of "Cloud Regimes Over the Amazon Basin: Perspectives From the GoAmazon2014/5 Campaign" by  Scott E. Giangrande, Dié Wang, and David B. Mechem

David K. Adams (dave.k.adams@gmail.com)

*General Comments*
I commend the authors for a well-written and interesting manuscript.  There is nothing that really needs to be modified in terms of the general structure.   One thing I think necessary is to provide a little more context for certain aspects of the study, such as the seasonality, diurnal cycle and the shallow-to-deep transition.  For example, the definition of wet, dry and transition seasons should be put in a bit more context of the literature.  Likewise the diurnal cycle.  Your definitions may be very "GOAmazon centric", I.e, the peculiarities of those 2 years.   Also, you mention the shallow-to-deep transition in passing, but this is actually a huge area of study in modeling as well as theory and observations.  And many of the shallow-to-deep transition  studies are based on Central Amazon conditions.  So I would include a little more context to indicate to the interested reader where they may explore more these important ideas.

*Specific Comments.*

**Introduction**

I think it is important to mention some of the previous work focusing on convection and surface atmosphere interactions as well as larger-scale forcing, such as WETAMC and TRMM-LBA. These studies provided data to examine another very important aspect of cloud development -- the inability of convective parameterizations  to capture something resembling a shallow-to-deep transition (Betts and Jakob 2002a,b).  And still to this day, the problem of the shallow-to-deep transition remains: what are the physical mechanisms (e.g., cold pool collisions, moisture preconditioning above the PBL, moisture flux convergence, etc…)?  And do models, even with convection "resolved", capture this transition in cloud development properly?    There is a very large literature on this (Khairoutdinov, M., and D. Randall (2006), Hohenegger, C., and B. Stevens, (2013); Wu, C.-M., B. Stevens, and A. Arakawa, (2009).  And many of these studies actually focus the Amazon.  This would go well with what you state in Lines 24-28 and with many  of the ideas you mention further on in the manuscript.

Line 34  Maybe write GCMs since otherwise it sounds like you are referring to a specific model called GCM.

Line 49-50   When you says "its" here you are referring to "changing nature of early transitions from dry and rainy seasons in the Amazon".  It's a little bit awkward, maybe you might want to rephrase this.

Line 83   "include estimates of"  sounds better to my ears.

Line 83 low- (surface to 3 km).  This seems very deep to me.   The PBL (over the forests nearby) is typically about 500 to 1000m, lower in the wet higher in the dry.   Can you give a little justification for choosing these layers?  From one of my student's thesis and our paper  (Lintner et al 2017),  if you told me, I can only choose one variable that exercises the most control over deep precipitating convection in the Central Amazon, I would say 700 to 500mb in terms of specific humidity.

Line 87  You should specify how you calculate CAPE, using a reversible or a pseudo-adiabatic and do you consider virtual temperature of the environmental profile in the vertical instead of just temperature. These can, in some conditions, may a pretty big difference.

Line 95  I would say "Amazonia (SIPAM) radar located on south end of Manaus" since I think the military base would be considered in Manaus.

Line 103-4.   I would disagree with this climatologically. The peak is typically around 2pm to 3pm, at least in terms of number of heaviest precipitation events.  It rained much less frequently in Manaus around 12pm, and usually less intense. See Adams et al. (2013) and also Ludmila's paper (Tanaka et al. 2016).   As I noted above, you may want to give a bit more context in terms of diurnal and seasonal cycles since they do vary a bit from study to study.   In Adams et al. 2013, and Lintner et al. 2017, as well as Adams et al 2015, we choose Jan-Apr as wet, May-June wet-to-dry transition, July-Sept dry and Oct-Dec as dry-to-wet.

Line 116   write " A simple cloud-type classification"

Line 130-135.  May give a little justification of these seasonal divisions.

Line 195 When you say the authors, you mean you? Maybe put "we" if that is what you mean.

Line 207-210.   I am a bit surprised by this.   I have found very little near surface humidity variability in the long-term Manaus sounding.  And I imagine over the rainforest, even less so, humidity is essentially constant and high.  Above the PBL, yes, there is where I think the important variability lies. The T3 site is an open pasture, I wonder if that accounts for lower atmo variability in water vapor.  The Manaus sounding is fairly close to the river, so maybe that is responsible for the small variability in lower levels.

Line 225   Definitely, April is rainy, it rains all the time and June is a transition month, not much rain, but cloudy a lot of the time (best time to visit Manaus for a tourist).

Line 238  Yes, I would say this is true, and there is very little lightning in January to April.  However, in Manaus, the locals call December "the lightning season".

Line 276  " as viewable by the current designations."  I think I know what you're saying, but it sound odd.

Line 291.  Large-scale w is a very tough variable to measure.  Strong positive w is associated with the in-cloud convective motions, as would be the latent heat release; however, the surrounding large scale atmosphere maybe still or subsiding.  Even if the atmosphere is perturbed by a gravity or Kelvin wave, w is very small.  As just an average measure of vertical motions it is o.k., but I wouldn't try to estimate variables like moisture flux convergence based on this estimate.

Line 345 - 355.   These are really a critical ideas about the STD that has implications far beyond this study.  So you might want to emphasize this point in your paper and consider a bit more of the literature even if just superficially.

Line 390 -394   Actually, see Figure 4 in Adams et al. 2015 that shows different behavior in the diurnal cycle of precipitable water vapor in and around Manaus.

Line 445-450  Yes, I would agree a lot of the convection in non-local forming most often to east of Manaus, and yes particularly during the transition and dry seasons.

**References**

Adams, D. K.,  Rui M. S. Fernandes, Kirk L. Holub, Seth I. Gutman, Henrique M. J. Barbosa, Luiz A. T. Machado, Alan J. P. Calheiros, Richard A. Bennett, E. Robert Kursinski, Luiz F. Sapucci, Charles DeMets, Glayson F. B. Chagas, Ave Arellano, Naziano Filizola, Alciélio A. Amorim Rocha**, Rosimeire Araújo Silva, Lilia M. F. Assunção, Glauber G. Cirino, Theotonio Pauliquevis, Bruno T. T. Portela, André Sá, Jeanne M. de Sousa, and Ludmila M. S. Tanaka, The Amazon Dense GNSS Meteorological Network: A New Approach for Examining Water Vapor and Deep Convection Interactions in the Tropics. *Bull. Amer. Meteor. Soc.*, **96**, 2151–2165.  doi: http://dx.doi.org/10.1175/BAMS-D-13-00171.1

Adams, D. K. , S. Gutman, K. Holub and D. Pereira, 2013: GNSS Observations of Deep Convective timescales in the Amazon, 2013:  *Geophysical Research Letter*s, **40**,1-6,doi:10.1002/grl.50573

Betts, A. K., and C. Jakob (2002), Study of diurnal convective precipitation over Amazonia using a single column model, J. Geophys Res., 107(D23),  4732, doi:10.1029/2001JD002264.

Betts, A. K., and C. Jakob , 2002b: Study of diurnal cycle of convective precipitation over Amazonia using a single column model.  J. Geophys. Res., 107, 4732, doi:10.1029/2002JD002264.

Hohenegger, C., and B. Stevens, 2013: Preconditioning deep convection with cumulus convection. J. Atmos. Sci., 70, 448–464, doi:10.1175/JAS-D-12-089.1

Khairoutdinov, M., and D. Randall (2006), High-resolution simulation of shallow-to-deep convection transition over land, J. Atmos. Sci., 63,3421–3436.

Lintner, B. R., D. K. Adams, K. A. Schiro,A. M. Stansfield, A. A. Amorim Rocha, and J. D. Neelin (2017), Relationships among climatological vertical moisture structure, column water vapor, and precipitation over the central Amazon in observations and CMIP5 models, Geophys. Res. Lett

Silva Dias MA, Rutledge MS, Kabat P, Silva Dias P, Nobre C, Fisch G, Dolman A, Zipser E, Garstang M, Manzi A, Fuentes J, Rocha H, Marengo J, Plana-Fattori A, S´a L, Alval´a R, Andreae M, Artaxo P, Gielow R, Gatti L.2002. Cloud and rain processes in a biosphere atmosphere I nteraction context in the Amazon region. Journal of Geophysical Research 107: 8072. DOI:10.1029/2001JD000335.

Tanaka, L. M. d. S., Satyamurty, P., & Machado, L. A. T. (2014). Diurnal variation of precipitation in central Amazon Basin. Int. J. Climatol., 34, 3574–3584. https://doi.org/10.1002/joc.3929

Wu, C.-M., B. Stevens, and A. Arakawa, 2009: What controls the transition from shallow to deep convection? J. Atmos. Sci., 66,1793–1806, doi:10.1175/2008JAS2945.1.

---

## Referee Comment (RC2) · Alan K. Betts (Referee) · 9 Mar 2020

Review of "Cloud Regimes Over the Amazon Basin: Perspectives From the GoAmazon2014/5 Campaign" by Scott E. Giangrande, Dié Wang, and David B. Mechem Alan K. Betts (akbetts@aol.com)

This data analysis from GOAmazon is valuable. An upfront discussion of the limitations in drawing conclusions from 2 years of data would be useful. The main challenge I had as an outside reviewer is unclear definitions early in the paper. I also suggest a change of style introducing each Figure would improve the readability. L84 LCL is missing from this list L92 Regime breakdowns (clusters) is not defined – see below

L99 Cluster routines incorporate: use of cluster is unclear L141 Finally you say: (Figure 1; Herein, we use the terms 'cluster' and 'regime' interchangeably). Looking back I see cluster is used in the abstract with no indication of what it is – derived from a model, described in section 2.2. The term is introduced in L 42-44. I recommend you rewrite L42-44 in the form We classify the primary thermodynamic regimes that are associated with the cloud observations over Manaus, based on a cluster analysis, by applying a k-means clustering technique (refs), to the morning radiosonde launches collected during the GoAmazon2014/5 campaign. This also isolates the potential controls of large-scale conditions on convective regimes. I find the use of 'breakdown' (as in L92 and elsewhere) confusing – perhaps because meteorologically it has been used for the breakdown of the dry season. Do you need it when you are simply describing the classification of days into regimes defined by the cluster analysis? Eg L149-51 could be written clearly as (consistent with L130): Figure 1 shows the cluster classification according to calendar-based Amazon definitions for the wet, dry and transitional seasons. The dry season months (Figure 1, bottom left panel) are predominantly associated with regimes 1-3, while the traditional Amazon wet season months (Figure 1, top right panel) are associated with regimes 4 and 5, with negligible contributions from the remaining regimes. Style. Generally I find texts much easier to read if each new Figure is always introduced with: Figure X shows... (rather than mentioned at the end of a sentence in parentheses) Figure 2 needs to reference Fig 1 for the cluster colors. Figure 11. How can the black plot for ALL be above all the regime classes at night? Isn't it an average of them?

PDF attached

Please also note the supplement to this comment:
https://www.atmos-chem-phys-discuss.net/acp-2020-67/acp-2020-67-RC2-supplement.pdf

---

## Referee Comment (RC3) · Yizhou Zhuang (Referee) · 27 Mar 2020

The authors present a k-means clustering analysis of thermodynamic conditions in the Central Amazon as represented by the radiosonde launches during the GOAmazon 2014/5 field campaign. They identified five regimes related to wet, transitional, and three dry types, respectively. Composite cloud and precipitation properties, convection statistics, large-scale circulation, and moisture advection related to these regimes are further contrasted. Finally, the authors discussed how these thermodynamic regimes can be linked to occurrence of different convection types. This manuscript is well written and it's interesting to see the clustering technique being applied to segregate local

thermodynamic controls as compared to simple seasonal composite analysis in most previous studies. Overall, most of their conclusions are consistent with previous research efforts, but they also provide another angle to understand the relationship between the complex convection characteristic over the Amazon Basin and various types of seasonal thermodynamic controls. However, I do have a few relatively minor comments, which are mostly related to clarification of some points and improvement of figures. After addressing these I think this manuscript should be ready for publication in ACP.

Major comments Selection of Radiosonde in Clear Conditions – section 2.1 In Line 99-100, the authors state that they only use radiosondes that launched in clear conditions. This is a very good practice for capturing pre-convection condition and studying shallow-to-deep convection transition. In Line 301-306, the authors also state that the enhanced moisture advection in regime 4 and 5 are not influenced by precipitation constrains since 1200UTC is prior to significant precipitation. However, I think the one-hour constraint for clear condition is probably too short to reduce the influence from early morning convection on the 1200 UTC sounding, especially during the wet and transition seasons. I would suggest use clear condition for at least 3-6 hours prior to sounding time and 1 hour after that, or at least discuss how the nocturnal convection can influence the 1200 UTC sounding and your results, especially those related to moisture. K-means Clustering Method – section 2.2 In Line 141-142, the authors described the clustering process as "radiosonde temperature and wind information is input at . . ." following Pope et al. 2009b. I think it can be made clearer that how many variables go into the clustering process (temperature, eastward and northward wind speed?)? Also, I'm concerned about why humidity information is not included as input since humidity is also a very important aspect in thermodynamics, and it can be very different during different seasons in Amazon and show significantly influence on buoyancy profile (e.g. Zhuang et al. 2017; 2018). I think justify this point will help readers better understand your basis for clustering. Also, I did not find out if the author preprocess the data before inputting them to k-means clustering. In Line 145-147,

"Although the authors prefer the solution that does not use normalized inputs . . . select consequences are discussed when these inputs result in divergent solutions". I'm still confused here, what kind of input is finally used to produce the final clustering results shown in the manuscript. I didn't find discussion about how this choice of input type would affect your results either. Perhaps it's better to move/add related discussion here or in the summary and discussion section. In addition, if you are using original or anomaly profile as input, did you assign weights for different variable? Would this affect your results? I'm asking this because these variables are in different units and the weighting can still have some influences even if the units are the same. Line 141-142, "input at 20 equally-spaced levels from the 1000 hPa to 200 hPa, . . .". I assume that this means equal weighting for different vertical layers. However, it seems to me the middle and upper level thermodynamics is much less important for convection than the lower troposphere. I'm wondering if some upper level thermodynamic disturbances could mask the lower level information and thus affect the clustering results. Maybe the authors can briefly comment this point. Also, the authors only show median profile in Figure 3, but I think a figure (either in the manuscript or the supplementals) showing both the mean/median and one standard deviation range of the input profiles in each regime would help address this point and show how well these five regimes represent the data. Large-Scale Synoptic Conditions – section 3.2 Please justify the use of 1000-hPa geopotential to represent large-scale circulation. For me, 1000-hPa is not a commonly used level for this kind of analysis, and I would prefer mean sea level pressure for surface system, 500-hPa or 200-hPa streamline for mid- to upper level circulation, 850-hPa wind for moisture advection analysis (consistent with many studies that low level moisture is more important for convection development and also your later results in Figure 7, 8 & 14). MCS in Regime 4 – section 4.3 Many studies (e.g. Williams et al. 2002, Zhuang et al. 2017) has shown that the transition season has a more unstable environment possibly contributed to its more intense convection than the wet season. It's also very interesting here (Line 441) to see that nearly half of the locally formed MCS are observed during the transitional regime. The authors

have compared the thermodynamics between the nonMCS and MCS cases in regime 4 (Line 454-461), but I'm more interested about why regime 4 can produce about twice MCS cases as many as those in regime 5. Can the results from early sections be used to explain this? Perhaps some of the discussions from Line 238-247 can be moved here. Summary and comparison of the results to broader literature – section 5 The summary section in the manuscript only lists some major findings throughout the previous results section. This section should include a more detailed discussion about how the results relate to and differ from previous studies. The bullet points for major finds should also be shortened to be simpler and more precise. It is also worth mentioning the advantage of applying this clustering technique to study thermodynamic controls of Amazon convection compared to regular seasonal composite analysis.

Minor comments Abstract Line 11: "three dry-season clusters". There are many places in the manuscript that use "dry season regime/cluster" or "drier season regime/cluster". I would suggest drop the "season" and simply use something like "dry regime/cluster" since these dry regime samples are also observed during the commonly defined wet or transition season. Also make sure the terminology is consistent throughout the manuscript. Line 12: ". . . for each regime for characteristic cloud frequency . . ." looks confusing. Please rephrase. Line 15: Again, what is "driest regimes". Is it just regime 1 or regime 1-3? Simply use "three dry regimes" if you were referring to regime 1-3? Line 15: What is "those" refer to? Line 15: "convective inhibition CIN". No need to write down abbreviation here. Section 2 Line 138-139: Please provide references for these commonly defined seasons. Line 141: "is input" → "are input" Line 142: "over North Australia" → "over the North Australia" Line 165-166: This sentence looks weird and hard to follow. Do you mean in their studies, rainfall trends and onset measures indicate 2014-2015 wet season onset occurred later? How can rainfall trend relate to onset time? Please rewrite and make it clearer. Section 3 Line 206: This information should be also included in the caption of Figure 4. Line 212: Is it 4-6 m/s in the dry season versus 2-4 m/s in the wet season? The dry regime spread looks wider than wet regime in Figure 4f. Line 267: "composite westerly wind components over the MAO

T3 site"? Where does this information come from? Figure 5e? I don't think the wind above the green star is significant westerlies. Line 267: Be consistent with site name. You used MAO site many times and MAO T3 a few times throughout the manuscript. Line 268: Same as the previous comment, I don't find the wind field above MAO in regime 4 much different from regime 5. Also, as I pointed out in the major comment, I would suggest use 850 hPa if you want to use wind to indicate moisture transport. This is more consistent with previous studies and your results in Figure 7-8. Section 4 Line 312: How is Figure 9 correspond to Figure 7? If there is no specific link, I think you can simply drop "that correspond to Figure 7". Line 353: "moister" → "wetter". Line 381: In Figure 11, why is the overall average rain rate higher than that of any regime during 03-12 UTC? Also, why is there no nocturnal precipitation here while there are significant clouds during late night and early morning in regime 3-5 in Figure 9. What is "regime-events having measurable precipitation"? Did you explain this before? Line 385-386: You mentioned the uncertainty of radar estimated precipitation here. Can you also briefly introduce in the method section how the precipitation is derived from radar reflectivity (Z-R relation)? As I can recall, they only use the wet season Z-R relationship from the disdrometer to calculate all precipitation data. This information can be found in the ARM-MAO PI dataset. Line 394: "the most frequent clouds we observe are" → "the time with most frequent clouds are" Line 399: "lower-relative domain rainfall rate"? Do you mean lower domain rain rate? Line 426: "e.g., defined by a minimum area of Z>20dBZ of <200km2". If this is the definition you used for non-precipitating event, remove "e.g.,". Also, perhaps "minimum area of Z>20dBZ is less than 200 km2" is better. minimal area of Z>20dBZ of 200 km2? What is definition of isolated, and widespread precipitation event. Line 431: "km2". Use superscript for square. Also check elsewhere in the manuscript. Figures Figure 1. Texts and numbers in this figure are too small. Consider increase the font size (also apply to some other figures), and use a legend like Figure 2 instead of listing R1, R2, . . . for all of the pie charts. Figure 2. Add a legend for different colors of dot in Figure 2a. Increase font size of R1, R2, . . . in Figure 2b. Also, to match the definition of seasons and make

it easier for readers to understand the result in Figure 2b, please consider only use four main color tones to represent wet, dry, and two transitional seasons. For different month in one season, just use different levels of darkness of the same color. Figure 4. Explain in the caption what's the thick black line in the middle of the density plot. Make the white number in bold font (also apply to Figure 12). Figure 7. "600-hPa/700-hPa" → "600-hPa (f-j) / 700-hPa (a-e)" Figure 8. What you plot is dash-dotted line not dashed line. Figure 9. Add unit to the colorbar. Why is tick numbers not aligned with the color? Figure 11. The shading areas look very narrow for standard deviation. Is it one standard deviation or standard error?

Please also note the supplement to this comment:
https://www.atmos-chem-phys-discuss.net/acp-2020-67/acp-2020-67-RC3-supplement.pdf

---

## Author Comment (AC1)

**Response to Reviewers: "Cloud Regimes Over the Amazon Basin: Perspectives From the GoAmazon2014/5 Campaign" by Scott E. Giangrande, Dié Wang, and David B. Mechem**

**Author Comments/Summary of Changes (Prepared by S. Giangrande):**

**We would like to thank the reviewers for their helpful comments and suggestions. Overall, we agree with the reviewers on these comments and have responded to and/or incorporated nearly all suggestions into the revised manuscript. In addition to our detailed responses to each reviewer, we call reviewer attention to additional cross-cutting changes:**

- **We improved manuscript language throughout, to also provide references for shallow-to-deep cloud transition topics, ideas that are later discussed by sections of the manuscript, and additional statements to caution potential readers on the limitations for this two-year dataset.**
- **Most manuscript figures have been modified for improved interpretation/clarity. New supplemental images provide larger-scale composite information at additional pressure levels (response to Reviewer 3).**
- **Several reviewers questioned our approach for select cumulative dataset plots, i.e., what was intended by 'ALL'/summary profiles? The 'all'-event behavior (includes every day, even those that did not match regime criteria, e.g., days having precipitation at 12 UTC). In general, we feel this way of including every event helps demonstrate whether cumulative regime properties missed significant cloud contributions from MCS.**

**Once again, we wish to thank the reviewers for their efforts in improving this manuscript. We hope our responses meet with reviewer approval. All reviewer comments are provided in this document.**

**Response to Reviewer 1**

*Review by David K. Adams (dave.k.adams@gmail.com)*

*General Comments*
*I commend the authors for a well-written and interesting manuscript. There is nothing that really needs to be modified in terms of the general structure. One thing I think necessary is to provide a little more context for certain aspects of the study, such as the seasonality, diurnal cycle and the shallow-to-deep transition. For example, the definition of wet, dry and transition seasons should be put in a bit more context of the literature. Likewise the diurnal cycle. Your definitions may be very "GOAmazon centric", I.e, the peculiarities of those 2 years. Also, you mention the shallow-to-deep transition in passing, but this is actually a huge area of study in modeling as well as theory and observations. And many of the shallow-to-deep transition studies are based on Central Amazon conditions. So I would include a little more context to indicate to the interested reader where they may explore more these important ideas.*

**We thank this reviewer for their insights, as well as their many helpful comments and suggestions. We have attempted to address/answer several of these concerns below, and make the appropriate changes for our revised manuscript.**

*Specific Comments. Introduction*

*I think it is important to mention some of the previous work focusing on convection and surface atmosphere interactions as well as larger-scale forcing, such as WETAMC and TRMM-LBA. These studies provided data to examine another very important aspect of cloud development -- the inability of convective parameterizations to capture something resembling a shallow-to-deep transition (Betts and Jakob 2002a,b). And still to this day, the problem of the shallow-to-deep transition remains: what are the physical mechanisms (e.g., cold pool collisions, moisture preconditioning above the PBL, moisture flux convergence, etc...)? And do models, even with convection "resolved", capture this transition in cloud development properly? There is a very large literature on this (Khairoutdinov, M., and D. Randall (2006), Hohenegger, C., and B. Stevens, (2013); Wu, C.-M., B. Stevens, and A. Arakawa, (2009).) And many of these studies actually focus the Amazon. This would go well with what you state in Lines 24-28 and with many of the ideas you mention further on in the manuscript.*

**Thank you for the comment. Although the shallow-to-deep transition is not central to our study, it remains an ongoing challenge, and we agree it is useful to improve our introduction to include the relevant shallow-to-deep issues that we touch on and that have been the focus**

of much previous research. This is an especially good idea because these themes are consistent with many GoAmazon motivations, ongoing GoAmazon activities, and DOE ARM activities proposed for upcoming AMF deployments.

*Line 34 Maybe write GCMs since otherwise it sounds like you are referring to a specific model called GCM.*

**Agree. Thanks.**

*Line 49-50 When you say "its" here you are referring to "changing nature of early transitions from dry and rainy seasons in the Amazon". It's a little bit awkward, maybe you might want to rephrase this.*

**We have reworded this portion of the introduction.**

*Line 83 "include estimates of" sounds better to my ears.*

**Agree. Fixed.**

*Line 83 low- (surface to 3 km). This seems very deep to me. The PBL (over the forests nearby) is typically about 500 to 1000m, lower in the wet higher in the dry. Can you give a little justification for choosing these layers? From one of my student's thesis and our paper (Lintner et al 2017), if you told me, I can only choose one variable that exercises the most control over deep precipitating convection in the Central Amazon, I would say 700 to 500mb in terms of specific humidity.*

**Agree. Thank you for the additional reference. We acknowledge that the approach was perhaps a bit arbitrary when selecting bulk layers of interest [e.g., 0--3 km, 3--6 km, 6--9 km, etc.], or choices of radiosonde thermodynamic parameters of interest. This was an attempt to provide options that may be of interest to multiple audiences, but in attempting to inform many, it is not well-aligned with any specific cloud/type. Initially, [0--3 km] was intended to coincide with the depth of the pre-convective cumulus layer, whereas the choice of the [3--6 km] layer was to gauge sensitivity of convection to mid-level RH / lapse rates. This approach was loosely aligned to previous studies, i.e., in their study of entrainment for cumulus congestus, Jensen and DelGenio (2006) use RH layers [2--4 km] and [5--7 km].**

**Jensen, M.P., and A.D. Del Genio, 2006: Factors limiting convective cloud-top height at the ARM Nauru Island climate research facility. J. Climate, 19, 2105-2117, doi:10.1175/JCLI3722.1.**

For reference (though not included in the revised manuscript), we have plotted the specific humidity (700 mb to 500 mb) in a manner similar to the other figures in the manuscript:

[Figure]

*Line 87 You should specify how you calculate CAPE, using a reversible or a pseudo-adiabatic and do you consider virtual temperature of the environmental profile in the vertical instead of just temperature. These can, in some conditions, may a pretty big difference.*

**Agree. We have attempted to correct inconsistencies with our statements on CAPE and its calculation. The section has been reworded. While there are differences among the different methods for calculating CAPE, we believe the relative relationships between the different regimes remains consistent with the descriptions we have provided in the text. Our assumptions follow a traditional parcel theory approach for CAPE calculation in Figure 4 and the values listed on the images in Figs 3 and 6:**

1. **Condensation/evaporation of water vapor only (and no ice phase at all);**
2. **Parcel uses irreversible ascent;**
3. **The virtual potential temperature framework is used [Bryan and Fritsch (2002)].**

**For demonstration purposes, the shaded areas on Figs 3/6 (CAPE) are calculated using air temperature, as according to Hobbs (1977); These are representative of traditional Skew-T plots.**

**Bryan, G.H. and J.M. Fritsch, 2002: A Benchmark Simulation for Moist Nonhydrostatic Numerical Models.** *Mon. Wea. Rev.,* **130, 2917–2928, https://doi.org/10.1175/1520-0493(2002)130 <2917:ABSFMN>2.0.CO;2**

**Hobbs, P. V., and J. M. Wallace, 1977: Atmospheric Science: An Introductory Survey. Academic Press, 350 pp.**

*Line 95 I would say "Amazonia (SIPAM) radar located on south end of Manaus" since I think the military base would be considered in Manaus.*

**Agree. Thanks.**

*Line 103-4. I would disagree with this climatologically. The peak is typically around 2pm to 3pm, at least in terms of number of heaviest precipitation events. It rained much less frequently in Manaus around 12pm, and usually less intense. See Adams et al. (2013) and also Ludmila's paper (Tanaka et al. 2016). As I noted above, you may want to give a bit more context in terms of diurnal and seasonal cycles since they do vary a bit from study to study. In Adams et al. 2013, and Lintner et al. 2017, as well as Adams et al 2015, we choose Jan-Apr as wet, May-June wet-to-dry transition, July-Sept dry and Oct-Dec as dry-to-wet.*

**Agree. We have modified this line. These statements were informed by the specifics of the GoAmazon deployment, where the timing of precipitation is influenced by several larger-scale, regional, local factors - as well as the limitations for a two-year sampling that may not be fully representative of climatology.**

*Line 116 write " A simple cloud-type classification"*

**Agree.**

*Line 130-135. May give a little justification of these seasonal divisions.*

**Agree. We have added some support to these.**

*Line 195 When you say the authors, you mean you? Maybe put "we" if that is what you mean.*

**Agree. We have changed this line to make it less confusing.**

*Line 207-210. I am a bit surprised by this. I have found very little near surface humidity variability in the long-term Manaus sounding. And I imagine over the rainforest, even less so, humidity is essentially constant and high. Above the PBL, yes, there is where I think the important variability lies. The T3 site is an open pasture, I wonder if that accounts for lower atmo variability in water vapor. The Manaus sounding is fairly close to the river, so maybe that is responsible for the small variability in lower levels.*

**Agree. This was a poor choice of wording. Here, we are referencing the variability in the mid-to-upper level moisture, and have clarified this in the revised text.**

*Line 225 Definitely, April is rainy, it rains all the time and June is a transition month, not much rain, but cloudy a lot of the time (best time to visit Manaus for a tourist).*

**Yes. March and April are very interesting months with some of the more impressive events ARM collected during the campaign, though these months are sometimes ignored in the literature that considers 'wet' season conditions (aka, studies that only include DJF or DJFM). Again, this possibly speaks to why calendar-driven studies could be deficient.**

*Line 238 Yes, I would say this is true, and there is very little lightning in January to April. However, in Manaus, the locals call December "the lightning season".*

**This is interesting to know. Thank you for providing the additional perspective. As before, this also speaks to the challenge when thinking of wet/transitional seasons in calendar terms as compared to those conditions conducive to storm electrification.**

*Line 276 " as viewable by the current designations." I think I know what you're saying, but it sound odd.*

**Agree. Modified.**

*Line 291. Large-scale w is a very tough variable to measure. Strong positive w is associated with the in-cloud convective motions, as would be the latent heat release; however, the surrounding large scale atmosphere maybe still or subsiding. Even if the atmosphere is perturbed by a gravity or Kelvin wave, w is very small. As just an average measure of vertical motions it is o.k., but I wouldn't try to estimate variables like moisture flux convergence based on this estimate.*

**Agree. The standard ARM single column model variational analysis products are provided as reference. However, we acknowledge that these products are sensitive to several factors including the precipitation fields (as noted in the manuscript), as well as use/location/number of radiosonde inputs. We have attempted to be in good communication with DOE/LLNL's group (under ARM Translator S. Xie, those who generate these products for DOE ARM) on issues related to these datasets, variability as based on presence/absence of particular inputs. We do emphasize that the large-scale vertical motion W and the horizontal convergence (and moisture convergence) are all mutually consistent and variationally constrained by the**

**observations and the mass continuity in the ECMWF model used in the variational analysis product.**

*Line 345 - 355. These are really a critical ideas about the STD that has implications far beyond this study. So you might want to emphasize this point in your paper and consider a bit more of the literature even if just superficially.*

**Agree. We have modified the text to acknowledge more of the existing literature on this topic, esp. some of the more recent studies coming out of GoAmazon2014/5.**

*Line 390 -394 Actually, see Figure 4 in Adams et al. 2015 that shows different behavior in the diurnal cycle of precipitable water vapor in and around Manaus.*

**Thank you for the additional reference.**

*Line 445-450 Yes, I would agree a lot of the convection in non-local forming most often to east of Manaus, and yes particularly during the transition and dry seasons.*

**Yes. As in our response to Reviewer 3, these MCS aspects are challenging to untangle. We suspect that follow-up deeper-dive 'regime 4' or similar activities would be beneficial -- our interests would be in refining which subsets of conditions within regime 4 promote MCS as compared to those conditions that do not.**

**Response to Reviewer 2**

*Review of "Cloud Regimes Over the Amazon Basin: Perspectives From the GoAmazon2014/5 Campaign" by Scott E. Giangrande, Dié Wang, and David B. Mechem*

*Alan K. Betts*

*This data analysis from GOAmazon is valuable. An upfront discussion of the limitations in drawing conclusions from 2 years of data would be useful. The main challenge I had as an outside reviewer is unclear definitions early in the paper. I also suggest a change of style introducing each Figure would improve the readability.*

**We thank the reviewer for their comments. We have reworked the introduction in parts (as also in response to Reviewer 1) and provide additional cautionary statements regarding the interpretation for the results from a 2-year dataset. We have also modified some of our text (i.e., in how we introduce figures) in response to the reviewer comments.**

*L84 LCL is missing from this list*

**Agree. Added.**

*L92 Regime breakdowns (clusters) is not defined – see below*
*L99 Cluster routines incorporate: use of cluster is unclear*
*L141 Finally you say: (Figure 1; Herein, we use the terms 'cluster' and 'regime' interchangeably). Looking back I see cluster is used in the abstract with no indication of what it is – derived from a model, described in section 2.2. The term is introduced in L 42-44. I recommend you rewrite L42-44 in the form*

> *We classify the primary thermodynamic regimes that are associated with the cloud observations over Manaus, based on a cluster analysis, by applying a k-means clustering technique (refs), to the morning radiosonde launches collected during the GoAmazon2014/5 campaign. This also isolates the potential controls of large-scale conditions on convective regimes.*

*I find the use of 'breakdown' (as in L92 and elsewhere) confusing – perhaps because meteorologically it has been used for the breakdown of the dry season. Do you need it when you are simply describing the classification of days into regimes defined by the cluster analysis?*
*Eg L149-51 could be written clearly as (consistent with L130):*

*Figure 1 shows the cluster classification according to calendar-based Amazon definitions for the wet, dry and transitional seasons. The dry season months (Figure 1, bottom left panel) are predominantly associated with regimes 1-3, while the traditional Amazon wet season months (Figure 1, top right panel) are associated with regimes 4 and 5, with negligible contributions from the remaining regimes.*

**We agree with the above comments. We have reworked several portions of our manuscript to make changes as recommended.**

*Style. Generally I find texts much easier to read if each new Figure is always introduced with: Figure X shows. . . (rather than mentioned at the end of a sentence in parentheses)*

**Agree. We have reworded several figure references to avoid (whenever possible) this way of introducing our figures.**

*Figure 2 needs to reference Fig 1 for the cluster colors.*

**Agree. We have modified several manuscript figures, as also in response to Reviewer 3. We have added this specific detail to the figure caption.**

*Figure 11. How can the black plot for ALL be above all the regime classes at night? Isn't it an average of them?*

**In "ALL" event profiles/curves, we include every day in the dataset. Thus, this also contains all days having precipitation at 1200 UTC, i.e., not an average of all 'regime' days. This was done to demonstrate whether summary regime properties (profiles) miss any significant cloud contribution (e.g., clouds associated with propagating/MCS events, etc.). Recall that in order for an event to be labelled as part of any 'regime', it cannot have precipitation / must be clear at the time of the morning radiosonde. We had concerns prior to submission that a small number of events (days) may have a significant impact on one or more summary cloud behaviors. Here, the primary change is that removing these 12 UTC rain/contaminated sonde events mostly removed a more continuous lower-level cloud condition over the site.**

**Response to Reviewer 3**

*The authors present a k-means clustering analysis of thermodynamic conditions in the Central Amazon as represented by the radiosonde launches during the GOAmazon 2014/5 field campaign. They identified five regimes related to wet, transitional, and three dry types, respectively. Composite cloud and precipitation properties, convection statistics, large-scale circulation, and moisture advection related to these regimes are further contrasted. Finally, the authors discussed how these thermodynamic regimes can be linked to occurrence of different convection types. This manuscript is well written and it's interesting to see the clustering technique being applied to segregate local thermodynamic controls as compared to simple seasonal composite analysis in most previous studies. Overall, most of their conclusions are consistent with previous research efforts, but they also provide another angle to understand the relationship between the complex convection characteristic over the Amazon Basin and various types of seasonal thermodynamic controls. However, I do have a few relatively minor comments, which are mostly related to clarification of some points and improvement of figures. After addressing these I think this manuscript should be ready for publication in ACP.*

**We thank the reviewer for their comments. We have attempted to address all of the reviewer comments (either to the revised manuscript, supplemental, or reviewer-only through these responses in limited examples). We hope our changes/responses will satisfy the reviewer.**

*Major comments Selection of Radiosonde in Clear Conditions – section 2.1*
*In Line 99-100, the authors state that they only use radiosondes that launched in clear conditions. This is a very good practice for capturing pre-convection condition and studying shallow-to-deep convection transition.*

**As the reviewer is aware, a challenge when using the 1800 UTC radiosondes is that once precipitating conditions are present in the domain (e.g., congestus, convection associated with cold-pools, etc.), there is far less certainty/confidence that the conditions (even when 'clear' of clouds/precipitation at the AMF T3 site) are not partially influenced by that ongoing convection.**

*In Line 301-306, the authors also state that the enhanced moisture advection in regime 4 and 5 are not influenced by precipitation constrains since 1200 UTC is prior to significant precipitation. However, I think the one- hour constraint for clear condition is probably too short to reduce the influence from early morning convection on the 1200 UTC sounding, especially during the wet and transition seasons. I would suggest use clear condition for at least 3-6 hours prior to sounding*

*time and 1 hour after that, or at least discuss how the nocturnal convection can influence the 1200 UTC sounding and your results, especially those related to moisture.*

**The reviewer correctly identifies the challenging interaction of scales in play in the Amazon environment, including the difficulty in cleanly separating the large-scale environment from the convection. The large-scale moisture advection will no doubt contain an influence from previous convective events, especially so for the regimes characterized by widespread precipitation and convective overturning (MCSs). Yet this is precisely the environment in which this convection forms. The best we can expect is to choose as our pre-convective periods those which minimize the immediate effects of convection, in particular cold pools. The 1200 UTC time seems to satisfy this ideal as close to possible, especially given the 6-hourly sounding interval.**

**For the benefit of the reviewer, we performed additional checks on these ideas (added notes, but opted not to make substantial changes to the manuscript). For the results presented in our study, approx. 93% of the radiosondes we labelled 'clear' were also 'clear' at 3 hrs prior (i.e., 0900 UTC to 1200 UTC) in the terms of surface precipitation (rain rate) measured by the ARM rain gauge. Most of these events were found scattered throughout the regimes (mostly regime 4 and 5), though we would suggest the regimes classified on those days were typically consistent with the surrounding days. Note also that simply removing these ~7% events would not imply a noticeable shift in current manuscript figures.**

*K-means Clustering Method – section 2.2*
*In Line 141-142, the authors described the clustering process as "radiosonde temperature and wind information is input at . . ." following Pope et al. 2009b. I think it can be made clearer that how many variables go into the clustering process (temperature, eastward and northward wind speed?)? Also, I'm concerned about why humidity information is not included as input since humidity is also a very important aspect in thermodynamics, and it can be very different during different seasons in Amazon and show significantly influence on buoyancy profile (e.g. Zhuang et al. 2017; 2018). I think justify this point will help readers better understand your basis for clustering. Also, I did not find out if the author preprocess the data before inputting them to k-means clustering.*

**Yes, this was a great catch by the reviewer! It is hugely important that we forgot to mention that 'dew point temperature' was also included (listed this as 'temperature'). The input variables that are included into this k-means methodology are: temperature, dew-point temperature, U, V.**

**As far as preprocessing the data: The radiosonde data are interpolated to a relatively fine grid (~2 hPa) prior to identifying the 20 equally-spaced levels from 1000 hPa to 200 hPa. These details have been checked to make this point more obvious in the revised manuscript.**

*In Line 145-147, "Although the authors prefer the solution that does not use normalized inputs . . . select consequences are discussed when these inputs result in divergent solutions". I'm still confused here, what kind of input is finally used to produce the final clustering results shown in the manuscript. I didn't find discussion about how this choice of input type would affect your results either. Perhaps it's better to move/add related discussion here or in the summary and discussion section. In addition, if you are using original or anomaly profile as input, did you assign weights for different variable? Would this affect your results? I'm asking this because these variables are in different units and the weighting can still have some influences even if the units are the same.*

**As in the responses above, there may be some confusion because we did not specify 'dew point temperature' as an input. Thus, we understand if the Reviewer was surprised by the results achieved without such information.**

**As to the reviewer comments thereafter: we did not employ scaled or similar sorts of inputs, e.g., 'normalized' or 'standardized'-type inputs, to generate the regime clusters for the results that we presented in the primary body of the manuscript. This is consistent with the previous Darwin studies (e.g., Pope et al.) that did not employ scaled inputs or similar to the best we can determine. However, we did perform testing for standardized inputs to determine if there may be improved or different results depending on the use of these forms of inputs. We indicate in our manuscript and its supplemental images that such preprocessing appears to increase the relative importance of the wind inputs in the cluster outcomes, e.g., relatively lessens the importance for temperature/dew-point inputs. This change seems to allow certain transitional regime differences to be noted. Nevertheless, certain regimes (that is, the most extreme wet and dry event days) are suggested as robust or consistent across our input tests.**

**Overall, we include some changes to the text, as was also associated with our original manuscript (Sec. 2.2, Sec 2.3). We hope these details/discussions are useful to the Reviewer and/or readers who may attempt similar or improved breakdowns. We have clarified the associated discussion on standardizing the inputs. Overall, we have found that these preprocessing efforts and potential sensitivities therein are not often well-communicated in the existing literature, and are sometimes non-obvious / not transparent.**

*Line 141-142, "input at 20 equally-spaced levels from the 1000 hPa to 200 hPa, . . .". I assume that this means equal weighting for different vertical layers. However, it seems to me the middle and upper level thermodynamics is much less important for convection than the lower troposphere. I'm wondering if some upper level thermodynamic disturbances could mask the lower level information and thus affect the clustering results. Maybe the authors can briefly comment this point. Also, the authors only show median profile in Figure 3, but I think a figure (either in the manuscript or the supplementals) showing both the mean/median and one standard deviation range of the input profiles in each regime would help address this point and show how well these five regimes represent the data.*

**This is an interesting comment. We tested the clusters to several factors that included different sets of inputs at different resolutions, e.g., whether 20-levels would faithfully reproduce the results we obtained from an entire radiosonde. When we include the full radiosonde or sub-sampling therein, there was not a significant change when compared to 20 coarsely-spaced levels. There are an infinite number of possibilities when considering how our intended audience may expand on these ideas to tackle specialized breakdowns. We agree with the reviewer that a focused set of inputs designed to explore the lowest-levels (as one example) may yield different results -- but, at the same time, we did not wish to unduly bias the regime identification with any preconceived notions such as minimizing the importance of middle-atmosphere stability. We do agree with the reviewer that such an effort (e.g., taking a deep dive into the regime 4 conditions to unravel MCS versus nonMCS) is potentially useful and something we wish to continue to explore when we begin idealized simulations on Amazon MCS events. One caution with such activities is that when one starts to fracture datasets into smaller subsets to interrogate for clues, there may not be sufficient data to perform efforts properly.**

**We have provided an image (below) for the benefit of the reviewer as a reference for the regime clusters as related to profile mean and spread of the conditions.**

[Figure]

*Large-Scale Synoptic Conditions – section 3.2 Please justify the use of 1000-hPa geopotential to represent large-scale circulation. For me, 1000-hPa is not a commonly used level for this kind of analysis, and I would prefer mean sea level pressure for surface system, 500-hPa or 200-hPa streamline for mid- to upper level circulation, 850-hPa wind for moisture advection analysis (consistent with many studies that low level moisture is more important for convection development and also your later results in Figure 7, 8 & 14).*

**This is a good comment. With ERA5 reanalysis, we are happy to provide 200hPa, 500hPa, and 850 hPa plots in the supplemental images (new Figures S5 - S7) to accompany the 1000 hPa plots in our manuscript. We feel the MSL pressure pattern is similar enough to the 1000 hPa geopotential heights that we have opted to retain the latter. Below is the regime breakdown for 850 hPa:**

[Figure]

*MCS in Regime 4 – section 4.3 Many studies (e.g. Williams et al. 2002, Zhuang et al. 2017) has shown that the transition season has a more unstable environment possibly contributed to its more intense convection than the wet season. It's also very interesting here (Line 441) to see that nearly half of the locally formed MCS are observed during the transitional regime. The authors have compared the thermodynamics between the nonMCS and MCS cases in regime 4 (Line 454-*

*461), but I'm more interested about why regime 4 can produce about twice MCS cases as many as those in regime 5. Can the results from early sections be used to explain this? Perhaps some of the discussions from Line 238-247 can be moved here.*

**Yes. We would also like to better understand this! One motivation was to identify the larger-scale controls associated with GoAmazon MCS events, an attempt to inform our idealized/real-world modeling activities (ongoing projects to simulate several GoAmazon MCS events). It was encouraging that 'regime 4' conditions (instead of 'wet' and 'transitional' conditions on calendar dates) were those that significantly favored MCS during this study. This finding was bolstered when we accounted for 'propagating' events (found in all regimes), but are not assumed to be 'locally' forced. Interestingly, 'propagating' MCS events under 'regime 4' suggested weaker conditions compared to MCS events initiating 'local', and weaker than the typical "non-MCS" regime 4 environment.**

**The most complex detail was our inability to identify differences in large-scale conditions in regime 4 that differentiated MCSs from non-MCS events, a finding that might greatly benefit potential parameterization development/evaluation. Expecting that all different convective modes would cleanly fall along thermodynamic and flow differences at a single point was rather ambitious (or naive?). It is logical that a more detailed analysis of 'regime 4' conditions, including additional observations, is likely required. We argue that these regimes mainly represent a possible starting point in understanding the variability of convective organization in the Amazon region. As the Reviewer notes, there was variability found within regime 4 that we highlighted for the readers -- (i.e., Lines 238-247, and associated materials) that suggested potential for stronger updrafts in the Oct-Nov windows. This shift toward stronger instability parameters seemed quite consistent with previous studies on storm electrification. However, mapping the intra-regime variability to MCS/nonMCS likelihood was not as straightforward with the parameters/quantities we considered.**

**Overall, regime 4 conditions include relatively high RH conditions and higher wind shear favorable to deep/organized convection. However, clearly other factors, beyond those calculated from the morning pre-convective radiosondes, control/regulate the development of organized convection.**

*Summary and comparison of the results to broader literature – section 5*
*The summary section in the manuscript only lists some major findings throughout the previ- ous results section. This section should include a more detailed discussion about how the results relate to and differ from previous studies. The bullet points for major finds should also be shortened to be simpler and more precise. It is also worth mentioning the advantage of applying this clustering*

*technique to study thermodynamic controls of Amazon convection compared to regular seasonal composite analysis.*

**Thank you for the comment. We have attempted to balance these reviewer comments, ideas with existing results, and references to the results of previous studies. Clearly, we agree with the reviewers that there are many recent GoAmazon studies in particular that we must acknowledge.**

*Minor comments*
*Abstract*
*Line 11: "three dry-season clusters". There are many places in the manuscript that use "dry season regime/cluster" or "drier season regime/cluster". I would suggest drop the "season" and simply use something like "dry regime/cluster" since these dry regime samples are also observed during the commonly defined wet or transition season. Also make sure the terminology is consistent throughout the manuscript.*

**We have attempted to modify some of these issues, also factoring the comments of Reviewer 2 regarding our use of 'cluster', 'breakdown', etc..**

*Line 12: ". . . for each regime for characteristic cloud frequency . . ." looks confusing. Please rephrase.*

**Ok.**

*Line 15: Again, what is "driest regimes". Is it just regime 1 or regime 1-3? Simply use "three dry regimes" if you were referring to regime 1-3?*

**Ok.**

*Line 15: What is "those" refer to? Line 15: "convective inhibition CIN". No need to write down abbreviation                                                                                                                          here.*

**Modified.**

*Section 2*
*Line 138-139: Please provide references for these commonly defined seasons. Line 141: "is input" → "are input"*

**Ok.**

*Line 142: "over North Australia" → "over the North Australia"*

**Modified.**

*Line 165-166: This sentence looks weird and hard to follow. Do you mean in their studies, rainfall trends and onset measures indicate 2014-2015 wet season onset occurred later? How can rainfall trend relate to onset time? Please rewrite and make it clearer.*

**Ok. Modified.**

*Section 3*
*Line 206: This information should be also included in the caption of Figure 4.*

**Ok.**

*Line 212: Is it 4-6 m/s in the dry season versus 2-4 m/s in the wet season? The dry regime spread looks wider than wet regime in Figure 4f.*

**Yes. Good catch. We made a mistake and reversed the seasons. Corrected to be more specific.**

*Line 267: "composite westerly wind components over the MAO T3 site"? Where does this information come from? Figure 5e? I don't think the wind above the green star is significant westerlies.*

**Ok. We have revised these comments to be more consistent.**

*Line 267: Be consistent with site name. You used MAO site many times and MAO T3 a few times throughout the manuscript.*

**Thanks. We will address this inconsistency.**

*Line 268: Same as the previous comment, I don't find the wind field above MAO in regime 4 much different from regime 5. Also, as I pointed out in the major comment, I would suggest use 850 hPa if you want to use wind to indicate moisture transport. This is more consistent with previous studies and your results in Figure 7-8.*

**We have added 850/500/200 hPa into the supplemental images. We think there is a relative shift in the lower-level winds between regimes 4 and 5 (as in our radiosonde composite and parameter images), as well as the overall regional patterns, but we agree that the visual differences in ERA5 patterns over the MAO between Regimes 4 and 5 are not obvious.**

*Section 4*
*Line 312: How is Figure 9 correspond to Figure 7? If there is no specific link, I think you can simply drop "that correspond to Figure 7".*

**Rephrased, as we agree this may be confusing.**

*Line 353: "moister" → "wetter".*

**Ok.**

*Line 381: In Figure 11, why is the overall average rain rate higher than that of any regime during 03-12 UTC? Also, why is there no nocturnal precipitation here while there are significant clouds during late night and early morning in regime 3-5 in Figure 9. What is "regime-events having measurable precipitation"? Did you explain this before?*

**As in response to Reviewer 2, our 'ALL' conditions were intended to represent all conditions. These include events that we excluded, e.g., because they did not satisfy our requirement for 'clear' conditions at 12 UTC. We wanted an 'all' behavior to help provide confidence that our regimes (or requirements) were not necessarily excluding too many MCS events (i.e., overnight, propagating) or other significant instances associated with nonclear conditions at 12 UTC.**

*Line 385-386: You mentioned the uncertainty of radar estimated precipitation here. Can you also briefly introduce in the method section how the precipitation is derived from radar reflectivity (Z-R relation)? As I can recall, they only use the wet season Z-R relationship from the disdrometer to calculate all precipitation data. This information can be found in the ARM-MAO PI dataset.*

**Ok. We have provided additional details. For domain averages from the VARANAL products, we do not believe this would represent a major change in our plots, but we have noted that these issues may suggest that 'drier' season behaviors are likely over-estimated (see below).**

**A more detailed answer is that our previous rainfall efforts from GoAmazon have looked at the differences between the seasons w/r/t Z-R relationships and sensitivities therein (crude**

bootstrapping, etc.), as found in Table 2 of Wang et al. (2018 ACP). These results were obtained using that same disdrometer (Parsivel) over the entire campaign period. Overall, the implication when thinking of the specific (contingent) regime breakdowns would be that the 'drier' season coefficients would arguably carry 'larger' (a-coefficient) for same (b-coefficient) in Z = aR^b forms. This implies that 'dry' season rainfall is associated with a larger Z for a given R. This makes physical sense insomuch that this implies larger drop sizes / less 'tropical'/'oceanic' behaviors than for the 'wetter' seasons.

The overall implication when interpreting the accumulation that is derived from a product that only uses 'wet' behaviors is that rainfall estimates will likely 'overestimate' rainfall rate and accumulation in the dry season (i.e, for the same Z, R should be smaller) if compared to estimates that used seasonally sensitive Z-R relationships. We have noted this in the revised manuscript.

Wang, D., Giangrande, S. E., Bartholomew, M. J., Hardin, J., Feng, Z., Thalman, R., and Machado, L. A. T.: The Green Ocean: precipitation insights from the GoAmazon2014/5 experiment, Atmos. Chem. Phys., 18, 9121–9145, https://doi.org/10.5194/acp-18-9121-2018, 2018.

*Line 394: "the most frequent clouds we observe are" → "the time with most frequent clouds are"*

**Ok.**

*Line 399: "lower-relative domain rainfall rate"? Do you mean lower domain rain rate?*

**Modified the text for clarity.**

*Line 426: "e.g., defined by a minimum area of Z>20dBZ of <200km2". If this is the definition you used for non-precipitating event, remove "e.g.,". Also, perhaps "minimum area of Z>20dBZ is less than 200 km2" is better. minimal area of Z>20dBZ of 200 km2? What is definition of isolated, and widespread precipitation event.*

**Ok. Fixed the corresponding text to improve clarity/interpretation.**

*Line 431: "km2". Use superscript for square. Also check elsewhere in the manuscript.*

**Ok.**

*Figures*

*Figure 1. Texts and numbers in this figure are too small. Consider increase the font size (also apply to some other figures), and use a legend like Figure 2 instead of listing R1, R2, . . . for all of the pie charts.*

**Ok. We added legends to Figures 1 and 13; Increased the font size for Figures 5, 6, 10, 13. Changes to Supplemental S1, S2 as well. Applied consistent changes to new supplemental figures as well.**

*Figure 2. Add a legend for different colors of dot in Figure 2a. Increase font size of R1, R2, . . . in Figure 2b. Also, to match the definition of seasons and make it easier for readers to understand the result in Figure 2b, please consider only use four main color tones to represent wet, dry, and two transitional seasons. For different month in one season, just use different levels of darkness of the same color.*

**Ok. We added the description of different colors in the Figure caption for Figure 2a, and we have changed the color of each month in Figure 2b (towards more of a wet-to-dry type of color scale that is less confusing than the previous rainbow colors).**

*Figure 4. Explain in the caption what's the thick black line in the middle of the density plot. Make the white number in bold font (also apply to Figure 12).*

**Ok.**

*Figure 7. "600-hPa/700-hPa" → "600-hPa (f-j) / 700-hPa (a-e)"*

**OK. Fixed.**

*Figure 8. What you plot is dash-dotted line not dashed line.*

**Thanks. Fixed.**

*Figure 9. Add unit to the colorbar. Why is tick numbers not aligned with the color?*

**Ok. Adjusted the colorbar and added the unit.**

*Figure 11. The shading areas look very narrow for standard deviation. Is it one standard deviation or standard error?*

**The shading areas are 1 sigma standard deviation.**